# Models Out Of Line:
# A Fourier Lens On Distribution Shift Robustness

**Sara Fridovich-Keil**[†,*], **Brian R. Bartoldson**[‡], **James Diffenderfer**[‡],
**Bhavya Kailkhura**[‡], **Peer-Timo Bremer**[‡]
[†]University of California, Berkeley   [‡]Lawrence Livermore National Laboratory

## Abstract

Improving the accuracy of deep neural networks on out-of-distribution (OOD) data is critical to an acceptance of deep learning in real world applications. It has been observed that accuracies on in-distribution (ID) versus OOD data follow a linear trend and models that outperform this baseline are exceptionally rare (and referred to as "effectively robust"). Recently, some promising approaches have been developed to improve OOD robustness: model pruning, data augmentation, and ensembling or zero-shot evaluating large pretrained models. However, there still is no clear understanding of the conditions on OOD data and model properties that are required to observe effective robustness. We approach this issue by conducting a comprehensive empirical study of diverse approaches that are known to impact OOD robustness on a broad range of natural and synthetic distribution shifts of CIFAR-10 and ImageNet. In particular, we view the "effective robustness puzzle" through a Fourier lens and ask how spectral properties of both models and OOD data correlate with OOD robustness. We find this Fourier lens offers some insight into why certain robust models, particularly those from the CLIP family, achieve OOD robustness. However, our analysis also makes clear that no known metric is consistently the best explanation of OOD robustness. Thus, to aid future research into the OOD puzzle, we address the gap in publicly-available models with effective robustness by introducing a set of pretrained CIFAR-10 models—*RobustNets*—with varying levels of OOD robustness.

## 1   Introduction

Deep learning (DL) holds great promise for solving difficult real-world problems in domains such as healthcare, autonomous driving, and cyber-physical systems. The major roadblock in adopting DL for real-world applications is that real-world data often deviates from the training data due to noise, corruptions, or other changes in distribution that may have temporal or spatial causes [3, 20, 41, 44]. DL models are known to be highly brittle under such distribution shifts, which raises the risk of making incorrect predictions with harmful consequences. Designing approaches to learn models that achieve both high accuracy on in-distribution data and high robustness to distribution shifts is paramount to the safe adoption of DL in real-world applications.

Several recent works [1, 35] have noticed an intriguing phenomenon: accuracy on in-distribution (ID) versus accuracy on distribution-shifted data obeys a linear trend across a wide range of models, corruption benchmarks, and prediction tasks. Unfortunately, as nearly all current DL models bear a substantial loss under distribution shifts, this linear relationship implies that our current training strategies are insufficient to bridge the gap between ID and out-of-distribution (OOD) performance. However, despite being exceedingly rare, models that break this linear trend to obtain effective robustness [1] exist—such models defy explanation by the wisdom that OOD accuracy and ID

---

[*]Corresponding author: `sfk@eecs.berkeley.edu`.

36th Conference on Neural Information Processing Systems (NeurIPS 2022).

accuracy should correlate strongly: they are "out of line". Identifying when and why such models arise is key to overcoming OOD brittleness in DL models.

Related research efforts have focused on improving OOD robustness via new techniques such as model pruning [8], data augmentation [5, 21], and ensembling large pre-trained models [37, 52]. While these efforts have created significant excitement in the ML robustness community, it remains unclear (a) under which conditions on OOD data these methods can improve effective robustness (ER), and (b) which underlying model properties make a DNN effectively robust. A more in-depth understanding of these phenomena is likely to help in bridging the gap between ID and OOD performance.

Towards achieving this goal, we carry out a comprehensive empirical study leveraging models trained using each of the aforementioned state-of-the-art approaches to improving OOD robustness. We analyze the impact of these robustness improvement methods on various model architectures, two clean datasets (CIFAR-10 and ImageNet), and a broad range of natural and synthetic distribution shifts. In particular, we view the OOD robustness puzzle through a Fourier lens and ask how spectral properties of OOD data and models influence their effective robustness. In this process, we design new metrics that capture Fourier sensitivity of models as test data moves farther away from the training data manifold. We make the following contributions:

- *A new perspective on the state of OOD robustness:* across diverse models, metrics, and distribution shifts, we observe that no one single metric rules them all, not even in-distribution accuracy. This accents the potential need for a multi-faceted approach to understanding OOD robustness, in contrast to ID generalization where single properties like model flatness can enjoy predictive success across an array of models [28].

- *Design of Fourier sensitivity metrics:* our new metrics quantify various model properties, including spectral ones that strongly correlate with the OOD robustness of ensembled CLIP models [52], explaining this robustness much better than ID accuracy.

- *A public collection of effectively robust models:* to help crack the OOD robustness puzzle, we make available (at https://github.com/sarafridov/RobustNets) a *RobustNets* dataset of CIFAR-10 models with effective robustness that stems from training under a variety of data augmentation and pruning schemes.

## 2  Related work

Distribution shift fragility is a well-documented phenomenon among neural networks [40, 41, 20, 19, 34] in which a model trained on an ID dataset performs markedly worse when evaluated on OOD data, even when humans find the OOD data just as easy to classify. These OOD accuracy gaps often follow a linear relationship between ID and OOD accuracy, but the slope and intercept of the linear trendline varies depending on the specific ID and OOD datasets [34, 15]. In our work, we focus on two popular image classification benchmark ID datasets, CIFAR-10 [30] and ImageNet [6], and an array of OOD benchmarks for each.

While most models follow the linear ID-OOD trendlines of Miller et al. [34], some models are "effectively robust" and appear above the trendline, with OOD accuracy higher than expected given ID accuracy [1]. These models are quite rare, and thus far have only been shown to arise through a few training paradigms: by pruning a model as a sort of regularization [8], by training with data augmentation intended to mimic the expected OOD test data [5, 21], or by pretraining on a larger and more diverse dataset and evaluating zero-shot [37], partially finetuning [1], or interpolating the weights of a zero-shot and a finetuned model [52].

Although as a community we are beginning to uncover robust training methods, it remains a mystery why these particular methods achieve robustness on particular distribution shifts. We study this mystery via a Fourier lens based on prior work involving both *image frequency* and *function frequency* analyses. Ortiz-Jimenez et al. [36], Yin et al. [55], and Sun et al. [47] study model sensitivity to Fourier perturbations of the input images and analyze how different data augmentations produce different Fourier sensitivities (and correspondingly robustness to image perturbations of different frequencies). Chen et al. [4] showed that exchanging amplitude and phase information between images (as a regularization scheme) has connections to robustness as well. Another line of research focuses on spectral bias [13, 2, 54, 38], wherein models prioritize learning low frequency

functions over the input space. This bias towards low function frequencies can also affect robustness and is influenced by training hyperparameters like data augmentation and weight decay [13].

Among the model properties we study as potential predictors of OOD accuracy are ID accuracy [34], model Jacobian norm [24], the linear pixel-space interpolation metrics of Fridovich-Keil et al. [13], and our own Fourier amplitude and phase interpolation metrics. We study these metrics over a wide range of robust and nonrobust models, including sparse models from [8] on CIFAR-10 and pretrained models from CLIP [37, 52] and RobustBench [5] on ImageNet.

## 3 Methods

**Datasets.** We consider two standard image classification tasks: CIFAR-10 [30] and ImageNet [6]. For each of these "in-distribution" (ID) datasets we consider a variety of "out of distribution" (OOD) datasets, both natural and synthetic. For CIFAR-10 we consider CIFAR-10.1 [40] and CIFAR-10-C [20]; for ImageNet we consider ImageNetV2 [41], ImageNet-C [19], ImageNet-R [23], ImageNet-A [9], ImageNet-Sketch [50], and ObjectNet [10].

**Models.** Our CIFAR-10 experiments use 3 different model architectures: Conv8 [12], ResNet18 [17], and VGG16 [45]. For each model architecture, we consider a range of model pruning strategies that have been studied for their effect on OOD robustness [8, 25] as some of these pruned models were demonstrated to have higher OOD robustness than dense models. Namely, lottery-ticket style pruning methods [12, 42, 39, 7] were able to provide robustness gains on CIFAR-10-C [8]. We outline the various pruning techniques and sparsity levels we utilized in detail in Section 4.1.

In our ImageNet experiments we use 28 standard (nonrobust) pretrained models from Torchvision [33], 5 pretrained models from the ImageNet-C leaderboard on RobustBench [5], and 33 models obtained by robustness-enhancing weight-space interpolation [52] between zero-shot and ImageNet-finetuned CLIP ViT-B/16, ViT-B/32, and ViT-L/14 [37]. More details on the ImageNet models we use from Torchvision and RobustBench can be found in Section A.1.

**Previously-proposed metrics: ID accuracy, Jacobian norm, pixel interpolation.** Because OOD robustness is a complex and multi-faceted problem, we make use of multiple previously-proposed model measurements and apply them to study effectively robust models. To the best of our knowledge, our work is the first application of most of these metrics (except for in-distribution accuracy) to the study of effective robustness. Specifically, we evaluate four previously-proposed metrics that have been observed to correlate with some notion of generalization: accuracy on unseen in-distribution data ("ID accuracy") [34], Jacobian norm [24], and within-class and between-class pixel interpolation high frequency fraction [13]. These latter metrics are analogous to the high frequency fractions we compute for amplitude and phase interpolation (described below), except that interpolating paths simply interpolate in pixel space between two images, where the two images are from either the same class ("within-class") or different classes ("between-class") [13]. Prior work observed that in-distribution generalization typically improves as within-class high frequency fraction decreases and between-class high frequency fraction increases [13]. Reducing the norm of the Jacobian of model outputs with respect to input data can push the decision boundary away from the training points, providing robustness to random perturbations of the input data [24]. Thus, models with smaller Jacobian norms may perform better on corrupted/shifted data. We estimate the norm of the Jacobian using a random-projection-based approach [24]; further details can be found in Section A.4.

**Fourier interpolation metrics.** We also introduce novel Fourier interpolation metrics to capture more information about model behavior that may help answer the robustness puzzle. Inspired by the pixel-space interpolation procedure of Fridovich-Keil et al. [13], we propose evaluating the smoothness of a model's predictions along image paths that perturb the Fourier amplitude or phase information of one test image towards another, while preserving the Fourier phase or amplitude (respectively) information of the original image. The intuition for this type of interpolation is that most of the semantic content of an image is contained within its Fourier phases, which encode structural information like edges. Accordingly, we can perturb the Fourier amplitude of an image without destroying its semantic meaning, and we can reliably destroy semantic content by perturbing Fourier phase. Example Fourier amplitude and phase interpolating paths on ImageNet are shown in Figure 1; example paths on CIFAR-10 and further computational details are included in Section A.3.

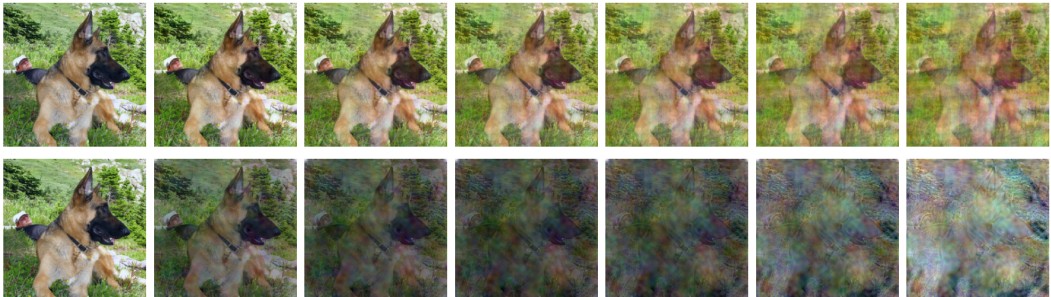

Figure 1: Example Fourier amplitude (top) and phase (bottom) interpolating paths from ImageNet. Each path includes 100 images; every 15th image is visualized here. The first image along each path is an unaltered image from the original validation set; the last image has the same Fourier phases (top) or amplitudes (bottom) as the original but has some of the Fourier amplitudes (top) or phases (bottom) of a random other image from the validation set. Amplitude interpolation preserves semantic content across usually the entire path, whereas phase interpolation destroys semantic content more rapidly. Note that the human visual system is robust to both types of corruption to some extent (see [51]).

More precisely, we compute an interpolating path by first randomly selecting two images $\mathbf{x}_0$ and $\mathbf{x}_1$ from the ID test/validation set. We perform standard preprocessing on each image, which involves cropping to a standard size (for ImageNet) and normalizing each pixel by the training mean and standard deviation (for both CIFAR-10 and ImageNet). We then compute the two-dimensional discrete Fourier transform (DFT) of each image and separate the complex-valued results into amplitude and phase components $\mathbf{a}_i$ and $\mathbf{p}_i$ for $i \in \{0, 1\}$. To construct an image along an amplitude interpolating path, we retain the original phase $\mathbf{p}_0$ and interpolate the low-frequency amplitude as $(1 - \lambda)\mathbf{a}_0 + \lambda\mathbf{a}_1$ (while preserving the high frequency amplitude), where $\lambda \in [0, 1]$, and produce an image via the inverse discrete Fourier transform. Phase interpolation is defined analogously; we retain the original amplitude and interpolate the low-frequency phase. We interpolate the lowest 40% of image frequencies for both amplitude and phase on CIFAR-10, the lowest 20% of image frequencies for phase on ImageNet, and all image frequencies for amplitude on ImageNet. We choose 100 evenly-spaced $\lambda$ values to define each path, and choose 5000 random paths for CIFAR-10 and 7000 for ImageNet. The images in Figure 1 are rescaled for visualization (essentially undoing the initial pixel-wise normalization).

We evaluate each model on each path, producing a probability vector over the available classes for each image along the path. We compute two metrics based on these path predictions: *high frequency fraction* (HFF) and *consistent distance* (CD). For high frequency fraction, we take the one-dimensional DFT of the 100 model predictions along the path, average the Fourier amplitudes among all the classes, and compute the fraction of the total amplitude above a frequency threshold (10% of the maximum frequency). This metric is always between 0 and 1 (usually between 0.1 and 0.3). The higher the average *high frequency fraction*, the more sensitive a model is to the given Fourier perturbation of its input images.

We also consider *consistent distance*, which is defined as the index of the first image along the path that the model classifies differently (by highest softmax prediction) than the original image. This metric is always between 1 and 100, and is often at least 40. Consistent distance is intended as a more intuitive metric to capture the same notion of robustness to corruptions of image Fourier content. In particular, Fourier amplitude corruptions change image statistics but do not hamper human semantic identification; a more robust model in this sense should have a *lower* high frequency fraction and a *higher* consistent distance on average compared to a less robust model.

## 4 Results

We are interested in understanding *when* (under what conditions on the distribution shift) and *why* (via model properties) neural networks can exhibit effective robustness. Although our experiments are correlational rather than causal, they provide evidence for and against various hypotheses regarding why effective robustness might emerge. We hope that future work will make use of our proposed

metrics and released model database *RobustNets* to answer the causal questions raised by our experiments, perhaps following Dziugaite et al. [11].

We perform a systematic study of three model interventions that have been shown to impact OOD robustness: pruning [8], data augmentation [5, 21], and weight ensembling [52]. For each intervention, we evaluate a benchmark set of pretrained CIFAR-10 and/or ImageNet models on the original in-distribution test set, several OOD test sets, and a suite of model property metrics capturing local smoothness (via model Jacobian norm), frequency response to pixel-space interpolation within and between classes [13], and Fourier amplitude and phase sensitivity (via *high frequency fraction* and *consistent distance*). Representative results from our analysis of these three interventions are presented in Sections 4.1, 4.2, 4.3, and A.10. For each dataset we consider a mixture of natural and synthetic distribution shifts: CIFAR-10.1 [40] and CIFAR-10-C [20], and ImageNet-V2 [41], ImageNet-R [23], ObjectNet [10], ImageNet-A [9], ImageNet-Sketch [50], and ImageNet-C [19]. Since CIFAR-10-C and ImageNet-C are each composed of 15 synthetic corruptions ranging from low to high frequency, we include results on representative corruptions of each frequency band in the main text and defer results on the remaining corruptions to the appendix. In all figures, we show 95% confidence intervals around each measurement (Clopper-Pearson for accuracy measurements and Gaussian bounds for averages of other metrics).

### 4.1 When and why does model pruning confer effective robustness?

Diffenderfer et al. [8] demonstrated the nuanced behavior that model pruning has on OOD robustness, specifically with respect to the distribution shift from CIFAR-10 to CIFAR-10-C. Notably, pruned models are capable of providing superior OOD robustness compared to dense, or unpruned, models. Furthermore, the effect, either positive or negative, and degree of robustness arising from model pruning is dependent on both the model architecture and the pruning algorithm. It remains unknown which properties of these pruned models contribute to their robustness on OOD data.

In an effort to better understand these findings, we further investigate this behavior by considering the three categories of pruning used in Diffenderfer et al. [8]: *traditional* [16, 57], *rewinding lottery-tickets* [12, 42], and *initialization lottery-tickets* [39, 7]. Note that we often abbreviate "lottery tickets" as LT. We provide details on each of the pruning techniques in Section A.2. For each pruning strategy and each architecture, we prune models to 50%, 60%, 80%, 90%, and 95% sparsity. For all pruning methods, pruning is performed in an unstructured (i.e., individual weights are pruned) and global manner (i.e., prune to a given sparsity across the entire network). Additionally, for traditional and initialization LTs, pruning was performed in a layerwise fashion (i.e., where each network layer is pruned to the given sparsity level). In Figure 2 we investigate *when* (on which types of distribution shifts) and in Table 1 we study *why* (via corresponding model property measurements) these differently-pruned CIFAR-10 models achieve their varying degrees of robustness.

Because different model architectures respond differently to different pruning methods, in each subplot of Figure 2 we color-code models by their architecture and fit a separate probit-domain OOD vs. ID accuracy regression line to each architecture, so that each regression captures a range of different pruning amounts (what fraction of weights are pruned). We display these regressions with opacity proportional to the fit quality $R^2$, so that regression lines that fit the data well are dark and easily seen, while lower-quality linear fits are more transparent. In the corresponding Tables 1, each reported slope ($m$) and $R^2$ value is the average of the corresponding values among the three architectures we consider. For ease of reading, we bold average $R^2$ values above 0.5, and among these entries we highlight the best (bright green) and second-best (faded green) $R^2$ values for each pruning method on each OOD dataset. Although these results are nuanced and raise additional open questions, we summarize a few key takeaways from our CIFAR-10 model pruning experiments:

- As reported in Miller et al. [34], ID accuracy is often a strong predictor of OOD accuracy; this trend holds across different pruning methods. However, the linear trend breaks down for mid and high frequency image corruptions like pixelate, Gaussian noise, and impulse noise.

- Even for low-frequency corruptions, different model architectures might lie on either the same or different robustness lines (as evidenced by comparing brightness and contrast).

- When trained without data augmentation, higher effective robustness is present on several mid- and high-frequency corruptions by non-residual architectures (i.e. Conv8 and VGG16) (see Figures 2

and 8). However, when training with data augmentation, less variation in effective robustness is realized by different architectures (see Figures 3 and 9).

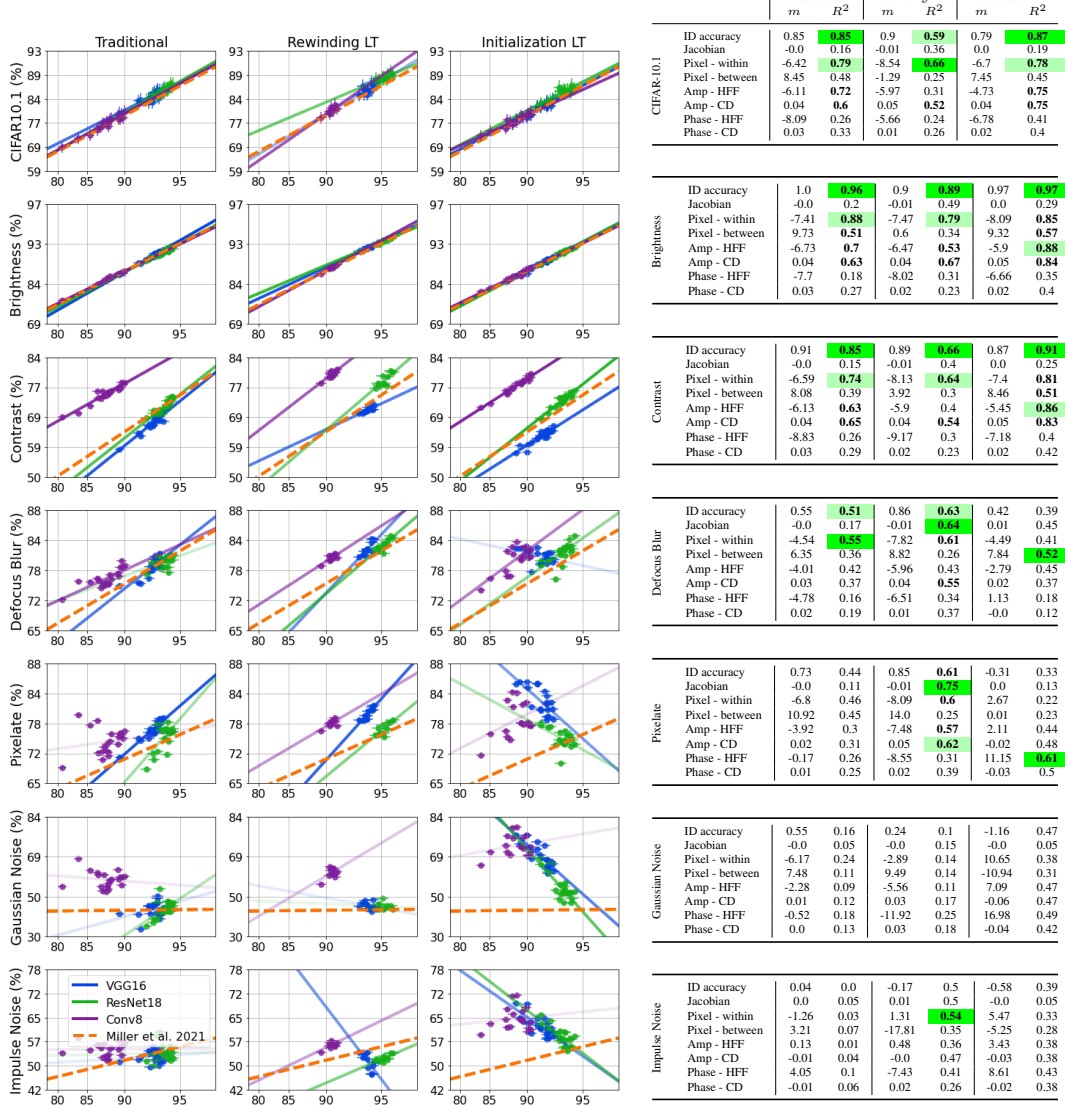

Figure 2: When does pruning affect OOD robustness?

Table 1: How well do metrics for pruned models correlate with OOD robustness?

|  |  | Traditional | | Rewinding LT | | Initialization LT | |
|---|---|---|---|---|---|---|---|
|  |  | $m$ | $R^2$ | $m$ | $R^2$ | $m$ | $R^2$ |
| CIFAR-10.1 | ID accuracy | 0.85 | **0.85** | 0.9 | **0.59** | 0.79 | **0.87** |
|  | Jacobian | -0.0 | 0.16 | -0.01 | 0.36 | 0.0 | 0.19 |
|  | Pixel - within | -6.42 | **0.79** | -8.54 | **0.66** | -6.7 | **0.78** |
|  | Pixel - between | 8.45 | 0.48 | -1.29 | 0.25 | 7.45 | 0.45 |
|  | Amp - HFF | -6.11 | **0.72** | -5.97 | 0.31 | -4.73 | **0.75** |
|  | Amp - CD | 0.04 | **0.6** | 0.05 | **0.52** | 0.04 | **0.75** |
|  | Phase - HFF | -8.09 | 0.26 | -5.66 | 0.24 | -6.78 | 0.41 |
|  | Phase - CD | 0.03 | 0.33 | 0.01 | 0.26 | 0.02 | 0.4 |
| Brightness | ID accuracy | 1.0 | **0.96** | 0.9 | **0.89** | 0.97 | **0.97** |
|  | Jacobian | -0.0 | 0.2 | -0.01 | 0.49 | 0.0 | 0.29 |
|  | Pixel - within | -7.41 | **0.88** | -7.47 | **0.79** | -8.09 | **0.85** |
|  | Pixel - between | 9.73 | 0.51 | 0.6 | 0.34 | 9.32 | 0.57 |
|  | Amp - HFF | -6.73 | **0.7** | -6.47 | 0.53 | -5.9 | **0.88** |
|  | Amp - CD | 0.04 | **0.63** | 0.04 | **0.67** | 0.05 | **0.84** |
|  | Phase - HFF | -7.7 | 0.18 | -8.02 | 0.31 | -6.66 | 0.35 |
|  | Phase - CD | 0.03 | 0.27 | 0.02 | 0.23 | 0.02 | 0.4 |
| Contrast | ID accuracy | 0.91 | **0.85** | 0.89 | **0.66** | 0.87 | **0.91** |
|  | Jacobian | -0.0 | 0.15 | -0.01 | 0.4 | 0.0 | 0.25 |
|  | Pixel - within | -6.59 | **0.74** | -8.13 | **0.64** | -7.4 | **0.81** |
|  | Pixel - between | 8.08 | 0.39 | 3.92 | 0.3 | 8.46 | 0.51 |
|  | Amp - HFF | -6.13 | **0.63** | -5.9 | 0.4 | -5.45 | **0.86** |
|  | Amp - CD | 0.04 | **0.65** | 0.04 | **0.54** | 0.05 | **0.83** |
|  | Phase - HFF | -8.83 | 0.26 | -9.17 | 0.3 | -7.18 | 0.4 |
|  | Phase - CD | 0.03 | 0.29 | 0.02 | 0.23 | 0.02 | 0.42 |
| Defocus Blur | ID accuracy | 0.55 | **0.51** | 0.86 | **0.63** | 0.42 | 0.39 |
|  | Jacobian | -0.0 | 0.17 | -0.01 | **0.64** | 0.01 | 0.45 |
|  | Pixel - within | -4.54 | **0.55** | -7.82 | **0.61** | -4.49 | 0.41 |
|  | Pixel - between | 6.35 | 0.36 | 8.82 | 0.26 | 7.84 | **0.52** |
|  | Amp - HFF | -4.01 | 0.42 | -5.96 | 0.43 | -2.79 | 0.45 |
|  | Amp - CD | 0.03 | 0.37 | 0.04 | **0.55** | 0.02 | 0.37 |
|  | Phase - HFF | -4.78 | 0.16 | -6.51 | 0.34 | 1.13 | 0.18 |
|  | Phase - CD | 0.02 | 0.19 | 0.01 | 0.37 | -0.0 | 0.12 |
| Pixelate | ID accuracy | 0.73 | 0.44 | 0.85 | **0.61** | -0.31 | 0.33 |
|  | Jacobian | -0.0 | 0.11 | -0.01 | **0.75** | 0.0 | 0.13 |
|  | Pixel - within | -6.8 | 0.46 | -8.09 | 0.6 | 2.67 | 0.22 |
|  | Pixel - between | 10.92 | 0.45 | 14.0 | 0.25 | 0.01 | 0.23 |
|  | Amp - HFF | -3.92 | 0.3 | -7.48 | **0.57** | 2.11 | 0.44 |
|  | Amp - CD | 0.02 | 0.31 | 0.05 | **0.62** | -0.02 | 0.48 |
|  | Phase - HFF | -0.17 | 0.26 | -8.55 | 0.31 | 11.15 | **0.61** |
|  | Phase - CD | 0.01 | 0.25 | 0.02 | 0.39 | -0.03 | 0.5 |
| Gaussian Noise | ID accuracy | 0.55 | 0.16 | 0.24 | 0.1 | -1.16 | 0.47 |
|  | Jacobian | -0.0 | 0.05 | -0.0 | 0.15 | -0.0 | 0.05 |
|  | Pixel - within | -6.17 | 0.24 | -2.89 | 0.14 | 10.65 | 0.38 |
|  | Pixel - between | 7.48 | 0.11 | 9.49 | 0.14 | -10.94 | 0.31 |
|  | Amp - HFF | -2.28 | 0.09 | -5.56 | 0.11 | 7.09 | 0.47 |
|  | Amp - CD | 0.01 | 0.12 | 0.03 | 0.17 | -0.06 | 0.47 |
|  | Phase - HFF | -0.52 | 0.18 | -11.92 | 0.25 | 16.98 | 0.49 |
|  | Phase - CD | 0.0 | 0.13 | 0.03 | 0.18 | -0.04 | 0.42 |
| Impulse Noise | ID accuracy | 0.04 | 0.0 | -0.17 | 0.5 | -0.58 | 0.39 |
|  | Jacobian | 0.0 | 0.05 | 0.01 | 0.5 | -0.0 | 0.05 |
|  | Pixel - within | -1.26 | 0.03 | 1.31 | **0.54** | 5.47 | 0.33 |
|  | Pixel - between | 3.21 | 0.07 | -17.81 | 0.35 | -5.25 | 0.28 |
|  | Amp - HFF | 0.13 | 0.01 | 0.48 | 0.36 | 3.43 | 0.38 |
|  | Amp - CD | -0.01 | 0.04 | -0.0 | 0.47 | -0.03 | 0.38 |
|  | Phase - HFF | 4.05 | 0.1 | -7.43 | 0.41 | 8.61 | 0.43 |
|  | Phase - CD | -0.01 | 0.06 | 0.02 | 0.26 | -0.02 | 0.38 |

- Initialization LT pruning confers robustness to high-frequency corruptions and can induce a negative correlation between ID and OOD accuracy for these corruptions.

- Jacobian norm is rarely a good predictor of OOD accuracy, but it does correlate negatively (lower norm is better) for models pruned by rewinding methods on mid-frequency corruptions.

- As reported in Fridovich-Keil et al. [13], lower high frequency fraction HFF (i.e. smoother predictions) for within-class pixel interpolation is a good predictor of OOD accuracy, at least for low-frequency distribution shifts where ID accuracy is also predictive.

- Amplitude and phase interpolation are sometimes predictive of OOD accuracy, particularly for initialization LT pruning, and in the expected direction of correlation (a better model should have lower HFF and higher CD). However, even these metrics fail to explain most of the robustness behavior for high (and some mid) frequency distribution shifts on CIFAR-10.

Please see Figure 10 for visualizations of the data fit associated with the most predictive metric for each distribution shift and pruning approach.

## 4.2 When and why does data augmentation confer effective robustness?

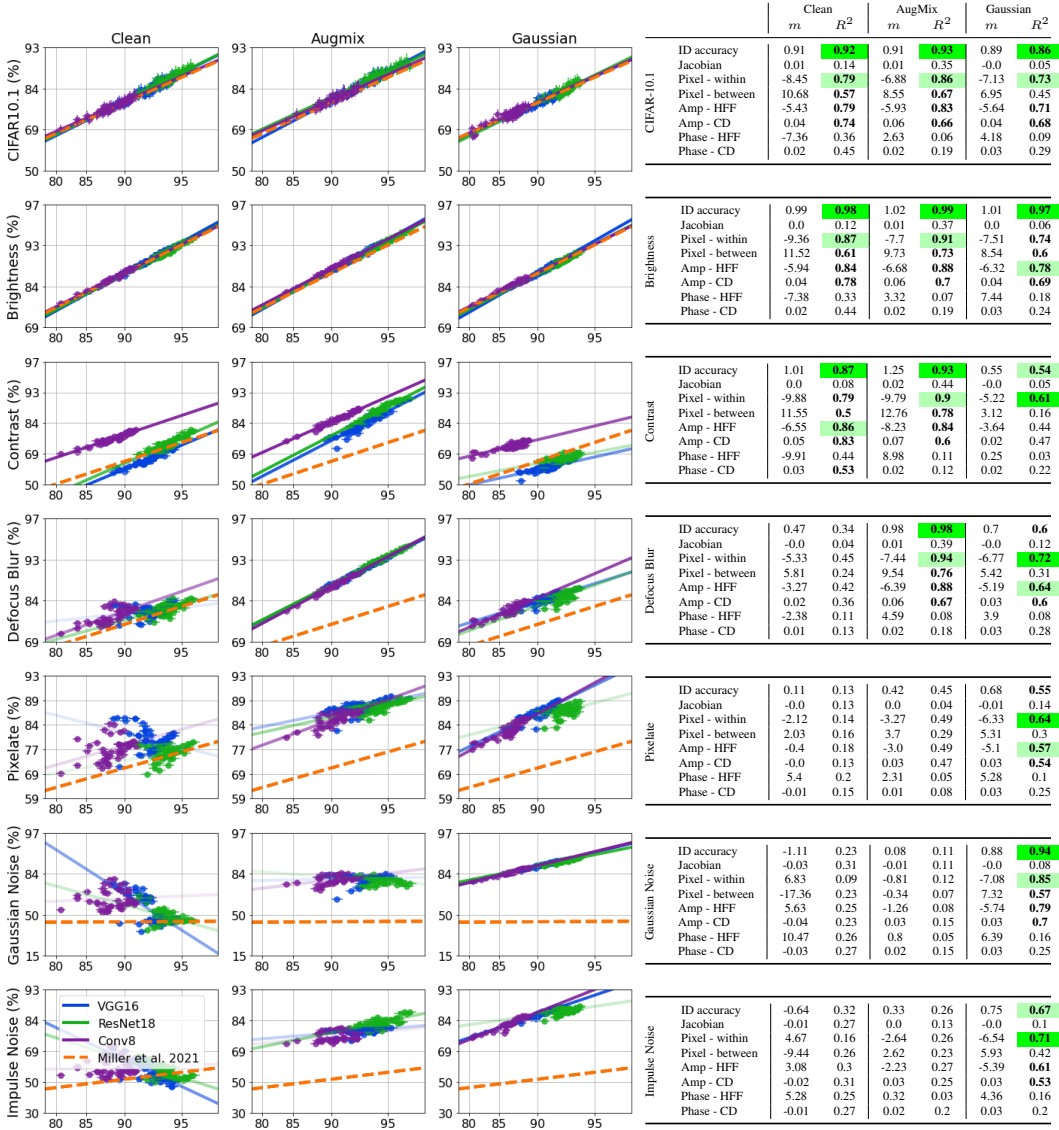

Figure 3: When does augmentation affect OOD robustness?

Table 2: How well do metrics for data-augmented models correlate with OOD robustness?

|  | Clean $m$ | Clean $R^2$ | AugMix $m$ | AugMix $R^2$ | Gaussian $m$ | Gaussian $R^2$ |
|---|---|---|---|---|---|---|
| **CIFAR-10.1** | | | | | | |
| ID accuracy | 0.91 | **0.92** | 0.91 | **0.93** | 0.89 | **0.86** |
| Jacobian | 0.01 | 0.14 | 0.01 | 0.35 | -0.0 | 0.05 |
| Pixel - within | -8.45 | **0.79** | -6.88 | **0.86** | -7.13 | **0.73** |
| Pixel - between | 10.68 | **0.57** | 8.55 | **0.67** | 6.95 | 0.45 |
| Amp - HFF | -5.43 | **0.79** | -5.93 | **0.83** | -5.64 | **0.71** |
| Amp - CD | 0.04 | **0.74** | 0.06 | **0.66** | 0.04 | **0.68** |
| Phase - HFF | -7.36 | 0.36 | 2.63 | 0.06 | 4.18 | 0.09 |
| Phase - CD | 0.02 | 0.45 | 0.02 | 0.19 | 0.03 | 0.29 |
| **Brightness** | | | | | | |
| ID accuracy | 0.99 | **0.98** | 1.02 | **0.99** | 1.01 | **0.97** |
| Jacobian | 0.0 | 0.12 | 0.01 | 0.37 | 0.0 | 0.06 |
| Pixel - within | -9.36 | **0.87** | -7.7 | **0.91** | -7.51 | **0.74** |
| Pixel - between | 11.52 | **0.61** | 9.73 | **0.73** | 8.54 | **0.6** |
| Amp - HFF | -5.94 | **0.84** | -6.68 | **0.88** | -6.32 | **0.78** |
| Amp - CD | 0.04 | **0.78** | 0.06 | **0.7** | 0.04 | **0.69** |
| Phase - HFF | -7.38 | 0.33 | 3.32 | 0.07 | 7.44 | 0.18 |
| Phase - CD | 0.02 | 0.44 | 0.02 | 0.19 | 0.03 | 0.24 |
| **Contrast** | | | | | | |
| ID accuracy | 1.01 | **0.87** | 1.25 | **0.93** | 0.55 | **0.54** |
| Jacobian | 0.0 | 0.08 | 0.02 | 0.44 | -0.0 | 0.05 |
| Pixel - within | -9.88 | **0.79** | -9.79 | **0.9** | -5.22 | **0.61** |
| Pixel - between | 11.55 | **0.5** | 12.76 | **0.78** | 3.12 | 0.16 |
| Amp - HFF | -6.55 | **0.86** | -8.23 | **0.84** | -3.64 | 0.44 |
| Amp - CD | 0.05 | **0.83** | 0.07 | **0.6** | 0.02 | 0.47 |
| Phase - HFF | -9.91 | 0.44 | 8.98 | 0.11 | 0.25 | 0.03 |
| Phase - CD | 0.03 | **0.53** | 0.02 | 0.12 | 0.02 | 0.22 |
| **Defocus Blur** | | | | | | |
| ID accuracy | 0.47 | 0.34 | 0.98 | **0.98** | 0.7 | **0.6** |
| Jacobian | -0.0 | 0.04 | 0.01 | 0.39 | -0.0 | 0.12 |
| Pixel - within | -5.33 | 0.45 | -7.44 | **0.94** | -6.77 | **0.72** |
| Pixel - between | 5.81 | 0.24 | 9.54 | **0.76** | 5.42 | 0.31 |
| Amp - HFF | -3.27 | 0.42 | -6.39 | **0.88** | -5.19 | **0.64** |
| Amp - CD | 0.02 | 0.36 | 0.06 | **0.67** | 0.03 | **0.6** |
| Phase - HFF | -2.38 | 0.11 | 4.59 | 0.08 | 3.9 | 0.08 |
| Phase - CD | 0.01 | 0.13 | 0.02 | 0.18 | 0.03 | 0.28 |
| **Pixelate** | | | | | | |
| ID accuracy | 0.11 | 0.13 | 0.42 | 0.45 | 0.68 | **0.55** |
| Jacobian | -0.0 | 0.13 | 0.0 | 0.04 | -0.01 | 0.14 |
| Pixel - within | -2.12 | 0.14 | -3.27 | 0.49 | -6.33 | **0.64** |
| Pixel - between | 2.03 | 0.16 | 3.7 | 0.29 | 5.31 | 0.3 |
| Amp - HFF | -0.4 | 0.18 | -3.0 | 0.49 | -5.1 | **0.57** |
| Amp - CD | -0.0 | 0.13 | 0.03 | 0.47 | 0.03 | 0.54 |
| Phase - HFF | 5.4 | 0.2 | 2.31 | 0.05 | 5.28 | 0.1 |
| Phase - CD | -0.01 | 0.15 | 0.01 | 0.08 | 0.03 | 0.25 |
| **Gaussian Noise** | | | | | | |
| ID accuracy | -1.11 | 0.23 | 0.08 | 0.11 | 0.88 | **0.94** |
| Jacobian | -0.03 | 0.31 | -0.01 | 0.11 | -0.0 | 0.08 |
| Pixel - within | 6.83 | 0.09 | -0.81 | 0.12 | -7.08 | **0.85** |
| Pixel - between | -17.36 | 0.23 | -0.34 | 0.07 | 7.32 | **0.57** |
| Amp - HFF | 5.63 | 0.25 | -1.26 | 0.08 | -5.74 | **0.79** |
| Amp - CD | -0.04 | 0.23 | 0.03 | 0.15 | 0.03 | **0.7** |
| Phase - HFF | 10.47 | 0.26 | 0.8 | 0.05 | 6.39 | 0.16 |
| Phase - CD | -0.03 | 0.27 | 0.02 | 0.15 | 0.03 | 0.25 |
| **Impulse Noise** | | | | | | |
| ID accuracy | -0.64 | 0.32 | 0.33 | 0.26 | 0.75 | **0.67** |
| Jacobian | -0.01 | 0.27 | 0.0 | 0.13 | -0.0 | 0.1 |
| Pixel - within | 4.67 | 0.16 | -2.64 | 0.26 | -6.54 | **0.71** |
| Pixel - between | -9.44 | 0.26 | 2.62 | 0.23 | 5.93 | 0.42 |
| Amp - HFF | 3.08 | 0.3 | -2.23 | 0.27 | -5.39 | **0.61** |
| Amp - CD | -0.02 | 0.31 | 0.03 | 0.25 | 0.03 | **0.53** |
| Phase - HFF | 5.28 | 0.25 | 0.32 | 0.03 | 4.36 | 0.16 |
| Phase - CD | -0.01 | 0.27 | 0.02 | 0.2 | 0.03 | 0.2 |

A common practice to protect against distribution shift is to train with data augmentation [5, 21]. Accordingly, we revisit each of the CIFAR-10 models considered in Section 4.1 with three different variations of data augmentation used during training: no augmentation (clean), AugMix [21], and Gaussian noise augmentation [29]. In Figure 3 we investigate *when* (on which types of distribution shifts) and in Table 2 we consider candidate explanations for *why* (via corresponding model property measurements) these differently-augmented CIFAR-10 models achieve varying degrees of robustness. As in Figure 2 and Table 1, we fit a separate regression line to each model architecture and average the resulting slope and $R^2$ values. The models that contribute to each regression line therefore share an architecture and an augmentation strategy, but vary by pruning method and amount. Our findings in this experiment echo those in Section 4.1, but we make the following additional observations:

- The closer the match between the training data and the OOD data in terms of their spectral profile (see Section A.5 for visualizations), the better ID accuracy, within-class pixel interpolation, and amplitude interpolation HFF and CD are at predicting OOD accuracy. This is evident from the high $R^2$ values for these metrics when predicting defocus blur accuracy for models trained with AugMix, and when predicting Gaussian or impulse noise accuracy for models trained with Gaussian noise.

- The closer the spectral match between the training data and the OOD data, the better the OOD accuracy. This unsurprising result is evident from the higher accuracies of AugMix-trained models on contrast and defocus blur corruptions and the higher accuracies of Gaussian-noise-trained models on Gaussian noise and impulse noise corruptions.

Please see Figure 11 for visualizations of the data fit associated with the most predictive metric for each distribution shift and data augmentation approach.

### 4.3   CLIP: Weight ensembling a robust model on ImageNet

Perhaps the most stunning example of OOD robustness for image classification is CLIP [37], which showed that a large model pretrained on a massive dataset and evaluated zero-shot on ImageNet achieves effective robustness on many OOD test sets simultaneously. Andreassen et al. [1] found that finetuning these robust models towards ImageNet improves ID accuracy on ImageNet but erodes effective robustness, and Wortsman et al. [52] offered a "best of both worlds" solution by interpolating in weight space between a pretrained zero-shot model and its finetuned counterpart, improving ID accuracy on ImageNet without sacrificing robustness. Accordingly, we study a set of 33 CLIP models obtained by interpolating the pretrained and finetuned versions of three CLIP ViT models (B/16, B/32, and L/14) [37, 52] with various relative weights on the two models. We compare this CLIP model set to a benchmark set of 28 standard (nonrobust) ImageNet pretrained models from Torchvision [33] as well as 5 pretrained models from the RobustBench [5] ImageNet-C leaderboard; these models typically use data augmentation to improve corruption performance but are not nearly as effectively robust as the CLIP models. We make the following observations in Figure 4 and Table 3:

- Although in-distribution accuracy is a reasonable predictor of OOD accuracy for most models and distribution shifts, it performs markedly worse when predicting OOD accuracy for CLIP models (and a couple of effectively robust RobustBench models on some OOD shifts).

- Fourier interpolation metrics for both amplitude and phase interpolation are strong predictors (often $R^2 \geq 0.9$) of OOD performance for CLIP models across OOD shifts. This suggests a hypothesis for *why* CLIP models are more robust: they are less sensitive to perturbations of Fourier amplitude and phase [51], which preserve image semantics for humans but confuse nonrobust models.

We also consider the ability of these model metrics to predict robustness in ImageNet models trained with various data augmentation approaches, which can produce effective robustness on many of the distribution shifts we consider [5]. Figures 5 and 13 show the most predictive metric's performance for these non-CLIP models from Figure 4. We again find that, although no single metric is best across all distribution shifts, our proposed Fourier interpolation metrics are often most predictive, outperforming in-distribution accuracy. See Appendix A.10 for full results and discussion.

## 5   Discussion

Our work is only the beginning of a full understanding of what makes models effectively robust to distribution shifts, starting with the domain of image classification where some models have demonstrated effective robustness. For example, Section 4.1 showed different types of model pruning can confer robustness to different distribution shifts, and while some of this robustness can be explained by improved behavior in response to amplitude and phase perturbations, much remains an open mystery.

As we saw in Section 4.2 (and has been noted by prior work [55, 21]), models can achieve robustness to some distribution shifts by training with data augmentations designed to imitate the expected OOD data. This approach can be a powerful way to mitigate expected test-time distribution shifts but leaves open the question of how to prepare for the unexpected. We learned in Section 4.3 that more robust CLIP models are less sensitive to perturbations of Fourier amplitude and phase, perhaps making them more attuned to human perceptions of semantic meaning. This finding provides an exciting opportunity for future work to confirm our results with both additional out-of-line models and causal

experiments; and to incorporate our Fourier interpolation metrics into training paradigms, for instance as regularizers or data augmentation strategies, to explicitly encode robustness in future models.

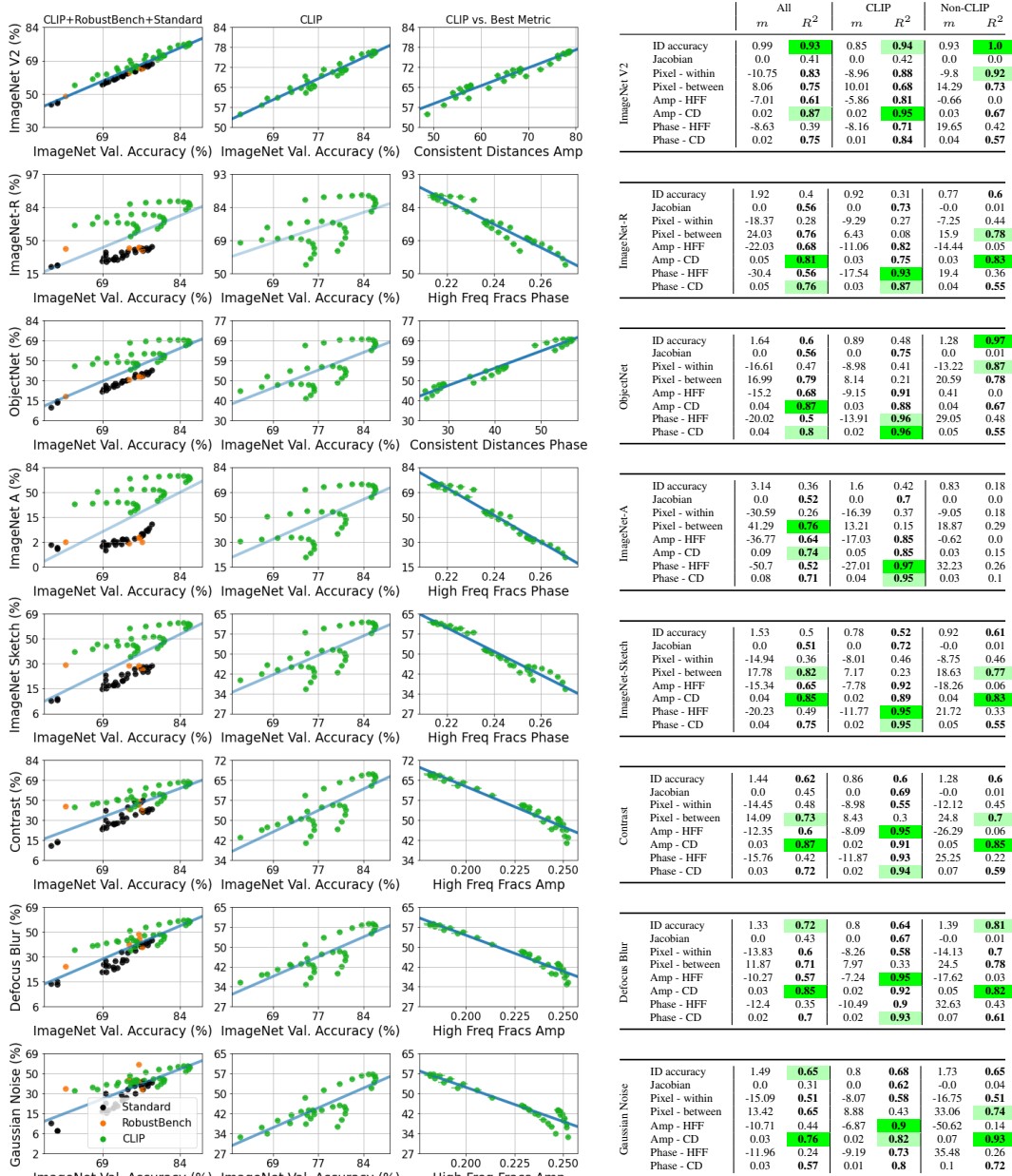

Figure 4: When does CLIP ensembling affect OOD robustness?

Table 3: How well do metrics for CLIP models correlate with OOD robustness?

| | | All | | CLIP | | Non-CLIP | |
|---|---|---|---|---|---|---|---|
| | | $m$ | $R^2$ | $m$ | $R^2$ | $m$ | $R^2$ |
| ImageNet V2 | ID accuracy | 0.99 | 0.93 | 0.85 | 0.94 | 0.93 | 1.0 |
| | Jacobian | 0.0 | 0.41 | 0.0 | 0.42 | 0.0 | 0.0 |
| | Pixel - within | -10.75 | 0.83 | -8.96 | 0.88 | -9.8 | 0.92 |
| | Pixel - between | 8.06 | 0.75 | 10.01 | 0.68 | 14.29 | 0.73 |
| | Amp - HFF | -7.01 | 0.61 | -5.86 | 0.81 | -0.66 | 0.0 |
| | Amp - CD | 0.02 | 0.87 | 0.02 | 0.95 | 0.03 | 0.67 |
| | Phase - HFF | -8.63 | 0.39 | -8.16 | 0.71 | 19.65 | 0.42 |
| | Phase - CD | 0.02 | 0.75 | 0.01 | 0.84 | 0.04 | 0.57 |
| ImageNet-R | ID accuracy | 1.92 | 0.4 | 0.92 | 0.31 | 0.77 | 0.6 |
| | Jacobian | 0.0 | 0.56 | 0.0 | 0.73 | -0.0 | 0.01 |
| | Pixel - within | -18.37 | 0.28 | -9.29 | 0.27 | -7.25 | 0.44 |
| | Pixel - between | 24.03 | 0.76 | 6.43 | 0.08 | 15.9 | 0.78 |
| | Amp - HFF | -22.03 | 0.68 | -11.06 | 0.82 | -14.44 | 0.05 |
| | Amp - CD | 0.05 | 0.81 | 0.03 | 0.75 | 0.03 | 0.83 |
| | Phase - HFF | -30.4 | 0.56 | -17.54 | 0.93 | 19.4 | 0.36 |
| | Phase - CD | 0.05 | 0.76 | 0.03 | 0.87 | 0.04 | 0.55 |
| ObjectNet | ID accuracy | 1.64 | 0.6 | 0.89 | 0.48 | 1.28 | 0.97 |
| | Jacobian | 0.0 | 0.56 | 0.0 | 0.75 | 0.0 | 0.01 |
| | Pixel - within | -16.61 | 0.47 | -8.98 | 0.41 | -13.22 | 0.87 |
| | Pixel - between | 16.99 | 0.79 | 8.14 | 0.21 | 20.59 | 0.78 |
| | Amp - HFF | -15.2 | 0.68 | -9.15 | 0.91 | 0.41 | 0.0 |
| | Amp - CD | 0.04 | 0.87 | 0.03 | 0.88 | 0.04 | 0.67 |
| | Phase - HFF | -20.02 | 0.5 | -13.91 | 0.96 | 29.05 | 0.48 |
| | Phase - CD | 0.04 | 0.8 | 0.02 | 0.96 | 0.05 | 0.55 |
| ImageNet-A | ID accuracy | 3.14 | 0.36 | 1.6 | 0.42 | 0.83 | 0.18 |
| | Jacobian | 0.0 | 0.52 | 0.0 | 0.7 | 0.0 | 0.0 |
| | Pixel - within | -30.59 | 0.26 | -16.39 | 0.37 | -9.05 | 0.18 |
| | Pixel - between | 41.29 | 0.76 | 13.21 | 0.15 | 18.87 | 0.29 |
| | Amp - HFF | -36.77 | 0.64 | -17.03 | 0.85 | -0.62 | 0.0 |
| | Amp - CD | 0.09 | 0.74 | 0.05 | 0.85 | 0.03 | 0.15 |
| | Phase - HFF | -50.7 | 0.52 | -27.01 | 0.97 | 32.23 | 0.26 |
| | Phase - CD | 0.08 | 0.71 | 0.04 | 0.95 | 0.03 | 0.1 |
| ImageNet-Sketch | ID accuracy | 1.53 | 0.5 | 0.78 | 0.52 | 0.92 | 0.61 |
| | Jacobian | 0.0 | 0.51 | 0.0 | 0.72 | -0.0 | 0.01 |
| | Pixel - within | -14.94 | 0.36 | -8.01 | 0.46 | -8.75 | 0.46 |
| | Pixel - between | 17.78 | 0.82 | 7.17 | 0.23 | 18.63 | 0.77 |
| | Amp - HFF | -15.34 | 0.65 | -7.78 | 0.92 | -18.26 | 0.06 |
| | Amp - CD | 0.04 | 0.85 | 0.02 | 0.89 | 0.04 | 0.83 |
| | Phase - HFF | -20.23 | 0.49 | -11.77 | 0.95 | 21.72 | 0.33 |
| | Phase - CD | 0.04 | 0.75 | 0.02 | 0.95 | 0.05 | 0.55 |
| Contrast | ID accuracy | 1.44 | 0.62 | 0.86 | 0.6 | 1.28 | 0.6 |
| | Jacobian | 0.0 | 0.45 | 0.0 | 0.69 | -0.0 | 0.01 |
| | Pixel - within | -14.45 | 0.48 | -8.98 | 0.55 | -12.12 | 0.45 |
| | Pixel - between | 14.09 | 0.73 | 8.43 | 0.3 | 24.8 | 0.7 |
| | Amp - HFF | -12.35 | 0.6 | -8.09 | 0.95 | -26.29 | 0.06 |
| | Amp - CD | 0.03 | 0.87 | 0.02 | 0.91 | 0.05 | 0.85 |
| | Phase - HFF | -15.76 | 0.42 | -11.87 | 0.93 | 25.25 | 0.22 |
| | Phase - CD | 0.03 | 0.72 | 0.02 | 0.94 | 0.07 | 0.59 |
| Defocus Blur | ID accuracy | 1.33 | 0.72 | 0.8 | 0.64 | 1.39 | 0.81 |
| | Jacobian | 0.0 | 0.43 | 0.0 | 0.67 | -0.0 | 0.01 |
| | Pixel - within | -13.83 | 0.6 | -8.26 | 0.58 | -14.13 | 0.7 |
| | Pixel - between | 11.87 | 0.71 | 7.97 | 0.33 | 24.5 | 0.78 |
| | Amp - HFF | -10.27 | 0.57 | -7.24 | 0.95 | -17.62 | 0.03 |
| | Amp - CD | 0.03 | 0.85 | 0.02 | 0.92 | 0.05 | 0.82 |
| | Phase - HFF | -12.4 | 0.35 | -10.49 | 0.9 | 32.63 | 0.43 |
| | Phase - CD | 0.02 | 0.7 | 0.02 | 0.93 | 0.07 | 0.61 |
| Gaussian Noise | ID accuracy | 1.49 | 0.65 | 0.8 | 0.68 | 1.73 | 0.65 |
| | Jacobian | 0.0 | 0.31 | 0.0 | 0.62 | -0.0 | 0.04 |
| | Pixel - within | -15.09 | 0.51 | -8.07 | 0.58 | -16.75 | 0.51 |
| | Pixel - between | 13.42 | 0.65 | 8.88 | 0.43 | 33.06 | 0.74 |
| | Amp - HFF | -10.71 | 0.44 | -6.87 | 0.9 | -50.62 | 0.14 |
| | Amp - CD | 0.03 | 0.76 | 0.02 | 0.82 | 0.07 | 0.93 |
| | Phase - HFF | -11.96 | 0.24 | -9.19 | 0.73 | 35.48 | 0.26 |
| | Phase - CD | 0.03 | 0.57 | 0.01 | 0.8 | 0.1 | 0.72 |

Our results provide a new perspective on the effective robustness problem by showing that no one single metric rules them all. This dictates the need for a multi-faceted analysis to understand OOD robustness, which is in sharp contrast to the in-distribution setting where a single property such as model flatness can reasonably explain generalization performance.

***RobustNets* benchmark.** The robustnesses of our analyzed CIFAR-10 models remain out-of-line. To track and aid progress on effective robustness prediction, we release our code as well as a database of our model weights and metric values at https://github.com/sarafridov/RobustNets. The models we study on ImageNet are publicly available.

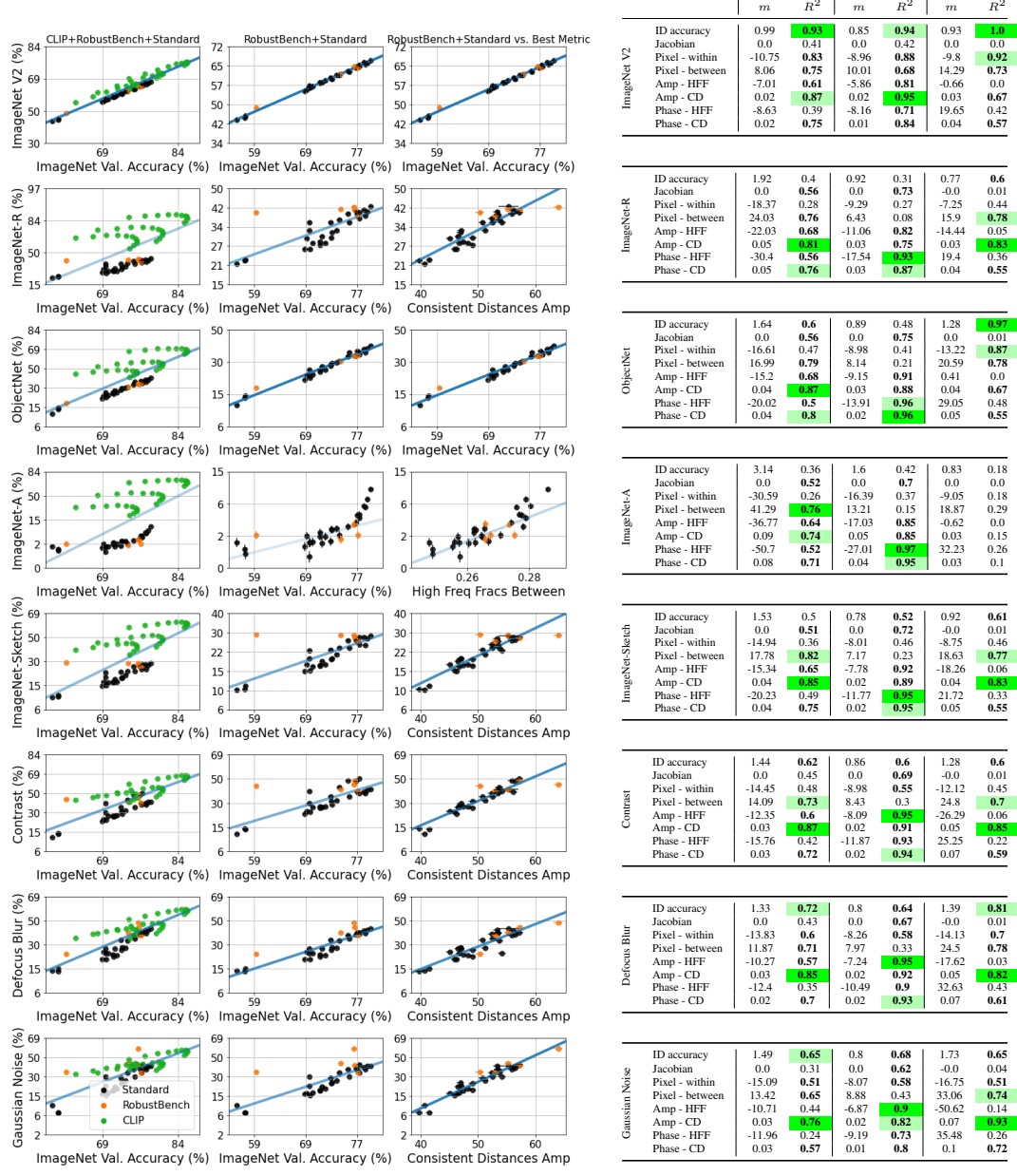

Figure 5: When does ImageNet data augmentation affect OOD robustness?

Table 4: How do metrics for ImageNet models correlate with OOD robustness?

| | | All | | CLIP | | Non-CLIP | |
|---|---|---|---|---|---|---|---|
| | | $m$ | $R^2$ | $m$ | $R^2$ | $m$ | $R^2$ |
| ImageNet V2 | ID accuracy | 0.99 | **0.93** | 0.85 | **0.94** | 0.93 | **1.0** |
| | Jacobian | 0.0 | 0.41 | 0.0 | 0.42 | 0.0 | 0.0 |
| | Pixel - within | -10.75 | **0.83** | -8.96 | 0.88 | -9.8 | **0.92** |
| | Pixel - between | 8.06 | **0.75** | 10.01 | **0.68** | 14.29 | **0.73** |
| | Amp - HFF | -7.01 | **0.61** | -5.86 | **0.81** | -0.66 | 0.0 |
| | Amp - CD | 0.02 | **0.87** | 0.02 | **0.95** | 0.03 | **0.67** |
| | Phase - HFF | -8.63 | 0.39 | -8.16 | **0.71** | 19.65 | 0.42 |
| | Phase - CD | 0.02 | **0.75** | 0.01 | **0.84** | 0.04 | 0.57 |
| ImageNet-R | ID accuracy | 1.92 | 0.4 | 0.92 | 0.31 | 0.77 | **0.6** |
| | Jacobian | 0.0 | **0.56** | 0.0 | **0.73** | -0.0 | 0.01 |
| | Pixel - within | -18.37 | 0.28 | -9.29 | 0.27 | -7.25 | 0.44 |
| | Pixel - between | 24.03 | **0.76** | 6.43 | 0.08 | 15.9 | **0.78** |
| | Amp - HFF | -22.03 | **0.68** | -11.06 | **0.82** | -14.44 | 0.05 |
| | Amp - CD | 0.05 | **0.81** | 0.03 | 0.75 | 0.03 | **0.83** |
| | Phase - HFF | -30.4 | **0.56** | -17.54 | **0.93** | 19.4 | 0.36 |
| | Phase - CD | 0.05 | **0.76** | 0.03 | **0.87** | 0.04 | **0.55** |
| ObjectNet | ID accuracy | 1.64 | **0.6** | 0.89 | 0.48 | 1.28 | **0.97** |
| | Jacobian | 0.0 | **0.56** | 0.0 | **0.75** | 0.0 | 0.01 |
| | Pixel - within | -16.61 | 0.47 | -8.98 | 0.41 | -13.22 | **0.87** |
| | Pixel - between | 16.99 | **0.79** | 8.14 | 0.21 | 20.59 | **0.78** |
| | Amp - HFF | -15.2 | **0.68** | -9.15 | **0.91** | 0.41 | 0.0 |
| | Amp - CD | 0.04 | **0.87** | 0.03 | **0.88** | 0.04 | **0.67** |
| | Phase - HFF | -20.02 | **0.5** | -13.91 | **0.96** | 29.05 | 0.48 |
| | Phase - CD | 0.04 | **0.8** | 0.02 | **0.96** | 0.05 | **0.55** |
| ImageNet-A | ID accuracy | 3.14 | 0.36 | 1.6 | 0.42 | 0.83 | 0.18 |
| | Jacobian | 0.0 | **0.52** | 0.0 | **0.7** | 0.0 | 0.0 |
| | Pixel - within | -30.59 | 0.26 | -16.39 | 0.37 | -9.05 | 0.18 |
| | Pixel - between | 41.29 | **0.76** | 13.21 | 0.15 | 18.87 | 0.29 |
| | Amp - HFF | -36.77 | **0.64** | -17.03 | **0.85** | -0.62 | 0.0 |
| | Amp - CD | 0.09 | **0.74** | 0.05 | **0.85** | 0.03 | 0.15 |
| | Phase - HFF | -50.7 | **0.52** | -27.01 | **0.97** | 32.23 | 0.26 |
| | Phase - CD | 0.08 | **0.71** | 0.04 | **0.95** | 0.03 | 0.1 |
| ImageNet-Sketch | ID accuracy | 1.53 | 0.5 | 0.78 | **0.52** | 0.92 | **0.61** |
| | Jacobian | 0.0 | **0.51** | 0.0 | **0.72** | -0.0 | 0.01 |
| | Pixel - within | -14.94 | 0.36 | -8.01 | 0.46 | -8.75 | 0.46 |
| | Pixel - between | 17.78 | **0.82** | 7.17 | 0.23 | 18.63 | **0.77** |
| | Amp - HFF | -15.34 | **0.65** | -7.78 | **0.92** | -18.26 | 0.06 |
| | Amp - CD | 0.04 | **0.85** | 0.02 | **0.89** | 0.04 | **0.83** |
| | Phase - HFF | -20.23 | 0.49 | -11.77 | **0.95** | 21.72 | 0.33 |
| | Phase - CD | 0.04 | **0.75** | 0.02 | **0.95** | 0.05 | **0.55** |
| Contrast | ID accuracy | 1.44 | **0.62** | 0.86 | **0.6** | 1.28 | **0.6** |
| | Jacobian | 0.0 | 0.45 | 0.0 | **0.69** | -0.0 | 0.01 |
| | Pixel - within | -14.45 | 0.48 | -8.98 | 0.55 | -12.12 | 0.45 |
| | Pixel - between | 14.09 | **0.73** | 8.43 | 0.3 | 24.8 | **0.7** |
| | Amp - HFF | -12.35 | **0.6** | -8.09 | **0.95** | -26.29 | 0.06 |
| | Amp - CD | 0.03 | **0.87** | 0.02 | **0.91** | 0.05 | **0.85** |
| | Phase - HFF | -15.76 | 0.42 | -11.87 | **0.93** | 25.25 | 0.22 |
| | Phase - CD | 0.03 | **0.72** | 0.02 | **0.94** | 0.07 | **0.59** |
| Defocus Blur | ID accuracy | 1.33 | **0.72** | 0.8 | **0.64** | 1.39 | **0.81** |
| | Jacobian | 0.0 | 0.43 | 0.0 | **0.67** | -0.0 | 0.01 |
| | Pixel - within | -13.83 | **0.6** | -8.26 | 0.58 | -14.13 | **0.7** |
| | Pixel - between | 11.87 | **0.71** | 7.97 | 0.33 | 24.5 | **0.78** |
| | Amp - HFF | -10.27 | **0.57** | -7.24 | **0.95** | -17.62 | 0.03 |
| | Amp - CD | 0.03 | **0.85** | 0.02 | **0.92** | 0.05 | **0.82** |
| | Phase - HFF | -12.4 | 0.35 | -10.49 | 0.9 | 32.63 | 0.43 |
| | Phase - CD | 0.02 | **0.7** | 0.02 | **0.93** | 0.07 | **0.61** |
| Gaussian Noise | ID accuracy | 1.49 | **0.65** | 0.8 | **0.68** | 1.73 | **0.65** |
| | Jacobian | 0.0 | 0.31 | 0.0 | **0.62** | -0.0 | 0.04 |
| | Pixel - within | -15.09 | **0.51** | -8.07 | 0.58 | -16.75 | **0.51** |
| | Pixel - between | 13.42 | **0.65** | 8.88 | 0.43 | 33.06 | **0.74** |
| | Amp - HFF | -10.71 | 0.44 | -6.87 | **0.9** | -50.62 | 0.14 |
| | Amp - CD | 0.03 | **0.76** | 0.02 | **0.82** | 0.07 | **0.93** |
| | Phase - HFF | -11.96 | 0.24 | -9.19 | **0.73** | 35.48 | 0.26 |
| | Phase - CD | 0.03 | **0.57** | 0.01 | **0.8** | 0.1 | **0.72** |

# Acknowledgments

We would like to thank Jonathan Frankle, Mitchell Wortsman, and Ludwig Schmidt for helpful discussion and pointers to resources. SFK acknowledges support from NSF GRFP. This work was performed under the auspices of the U.S. Department of Energy by Lawrence Livermore National Laboratory under Contract DE-AC52-07NA27344 and was supported by the LLNL-LDRD Program under Project No. 2022-SI-004 (LLNL-CONF-835574).

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
