# A Appendix

## A.1 ImageNet Model Details

We use the following models from Torchvision (details at `https://pytorch.org/vision/0.8/models.html`): `alexnet` [31], `vgg11` [46], `vgg11_bn`, `vgg13`, `vgg13_bn`, `vgg16`, `vgg16_bn`, `vgg19`, `vgg19_bn`, `resnet18` [18], `resnet34`, `resnet50`, `resnet101`, `resnet152`, `squeezenet1_0` [27], `squeezenet1_1`, `densenet121` [26], `densenet169`, `densenet161`, `densenet201`, `googlenet` [48], `shufflenet_v2_x1_0` [32], `mobilenet_v2` [43], `resnext50_32x4d` [53], `resnext101_32x8d`, `wide_resnet50_2` [56], `wide_resnet101_2`, and `mnasnet1_0` [49].

We use the following models from the RobustBench [5] model zoo on ImageNet corruptions (details at `https://github.com/RobustBench/robustbench#model-zoo`): `Geirhos2018_SIN` [14], `Geirhos2018_SIN_IN`, `Geirhos2018_SIN_IN_IN`, `Hendrycks2020Many` [22], and `Hendrycks2020AugMix` [21].

From the CLIP family [37], we use 33 models formed by weight-space interpolation between zero-shot and finetuned ViT-B/16, ViT-B/32, and ViT-L/14 (11 models from each architecture).

## A.2 Pruning methods

In Section 4.1 we present results considering three categories of model pruning used in Diffenderfer et al. [8]: *traditional* (fine-tuning [16], gradual magnitude pruning [57]), *rewinding lottery-tickets* (weight-rewinding [12], learning-rate rewinding [42]), and *initialization lottery-tickets* (edgepopup [39], biprop [7]). Here we provide a brief description of each pruning method.

**Traditional.** As the name suggests, fine-tuning prunes a model at the end of the regular training period by removing $p\%$ of the weights with the smallest magnitude, then fine tunes the remaining weights using the learning rate at the end of the regular training period. Gradual magnitude pruning progressively removes weights during training in accordance with a sparsity scheduler (see Fig. 1 in Zhu and Gupta [57]) until the target sparsity is reached. Typically, some training takes place before and after pruning in gradual magnitude pruning.

**Rewinding lottery tickets.** Rewinding lottery tickets perform repeated training-then-pruning steps in which some percentage of the remaining weights are pruned at each pruning step (traditionally 20% of the remaining weights are pruned at each step). After each training-then-pruning step, either the weights and learning rate are rewound to their values at initialization (i.e. weight-rewinding) or the learning rate is rewound to a value earlier in the training process (learning-rate rewinding). The first rewinding lottery ticket method supported the Lottery Ticket Hypothesis [12] which suggests that sparse subnetworks exist within randomly initialized dense networks that can be trained to the same accuracy as dense networks.

**Initialization lottery tickets.** Stronger versions of this hypothesis, such as the Strong [39] and Multi-Prize Lottery Ticket Hypotheses [7] suggest that such subnetworks exist at initialization that achieve comparable accuracy to the dense network without requiring any training and, further, that these subnetworks' weights can be binarized. Edgepopup identifies such subnetworks using surrogate scores to identify the most important weights at initialization. Biprop integrates weight and/or activation binarization into the search process for such subnetworks resulting in binary-weight or binary-weight and activation networks (BNNs). Note that biprop networks in this paper only make use of binarized weights.

## A.3 Fourier interpolation on CIFAR-10 and computational details

Figure 1 in the main paper shows example Fourier amplitude and phase interpolating paths on ImageNet. Figure 6 shows example Fourier amplitude and phase interpolating paths on CIFAR-10, where again the images are rescaled for visualization (undoing the pixel-wise normalization).

Relatedly, note that in Section 3, the "lowest 40% of image frequencies" we reference is with respect to the maximum spatial frequency of the image (a function of the pixel resolution) and not based on

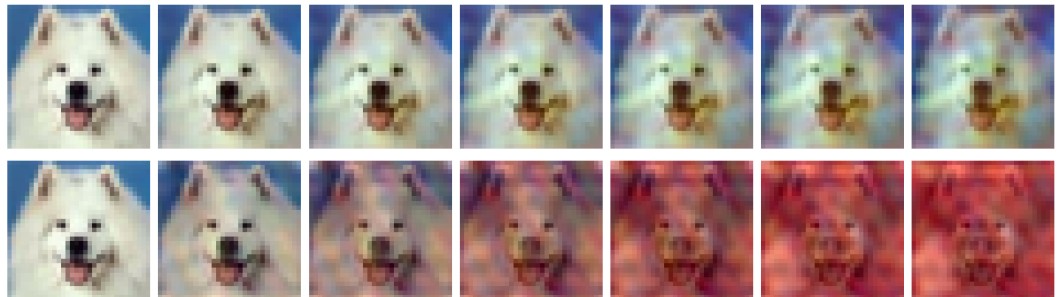

Figure 6: Example Fourier amplitude (top) and phase (bottom) interpolating paths from CIFAR-10. Each path includes 100 images; every 15th image is visualized here. The first image along each path is an unaltered image from the original test set; the last image has the same Fourier phases (top) or amplitudes (bottom) as the original but has some of the Fourier amplitudes (top) or phases (bottom) of a random other image from the validation set. Amplitude interpolation produces a corruption that preserves semantic content, whereas phase interpolation destroys semantic content.

the actual power spectral density of the image. Further, the interpolation is done using a square mask on the DFT of the image, which interpolates the lower image frequencies but leaves the higher image frequencies unperturbed.

### A.4  Jacobian norm computational details

We estimate the norm of the Jacobian using a random-projection-based approach that estimates the Jacobian norm as a function of a batch of samples of size $B$ and $n_{proj}$ projection vectors [24]. For CIFAR-10 and ImageNet, we set $n_{proj}$ to 10 and 50, respectively; we set $B$ to 400 and 1000, respectively. Error bars for our estimates are 95% confidence intervals constructed under the assumption that the $n_{proj}B$ unique Jacobian norm estimates made by the projection method are i.i.d., which is consistent with the error order of $(n_{proj}B)^{-\frac{1}{2}}$ expected by Hoffman et al. [24] when $B \gg 1$. Under this assumption, which may lead to underestimation of the true error if violated, our choice of $n_{proj}$ and $B$ ensures that the average error bar is less than 2% of the size of the estimated Jacobian norm (and all error bars are less than 8% of the size of the estimated Jacobian norm). Our estimated Jacobian norms are between 60 and 2300 for ImageNet models and between 2 and 50 for CIFAR-10 models.

### A.5  Power spectral density of distribution shifts

To explore the behavior of models on OOD data, it is helpful to characterize the nature of the OOD data with respect to the ID test data; indeed this type of characterization is common practice [55]. Specifically, considering the difference in power spectral density (PSD) between each of the OOD test sets and the original ID test set allows us to categorize each OOD test set as being concentrated on low, mid, or high frequency perturbations with respect to the ID test set. We perform this characterization for CIFAR-10.1 and each of the 15 corruptions in CIFAR-10-C.

The PSDs shown in Figure 7 reflect the distribution *shift* between each OOD dataset and the original CIFAR-10 test set, and are sorted roughly in order of increasing frequency. Of note, this PSD characterization illustrates that the OOD shift encountered on CIFAR-10.1 data is composed of low-frequency information, much like the brightness and contrast CIFAR-10-C corruptions.

As each of the CIFAR-10-C corruptions are transformations applied to the original CIFAR-10 test images, the PSD for each CIFAR-10-C corruption is computed by taking the difference between each corrupted and original test image, computing the PSD of the resulting difference image, then averaging the PSDs for all images of the given corruption. As CIFAR-10.1 images are not corruptions of the original CIFAR-10 test images, the PSD characterization must be computed in a different manner. First, for each of the ten CIFAR-10 classes we compute the average PSD for CIFAR-10 test images and CIFAR-10.1 images and take the difference of these class PSDs. The final approximated PSD representing the CIFAR-10.1 shift is taken to be the average of the difference PSDs for the ten classes.

The dramatic variations in frequency content among these different distribution shifts underscore the difficulty of the robustness puzzle; models may need different tools or properties to achieve robustness on different types of distribution shifts.

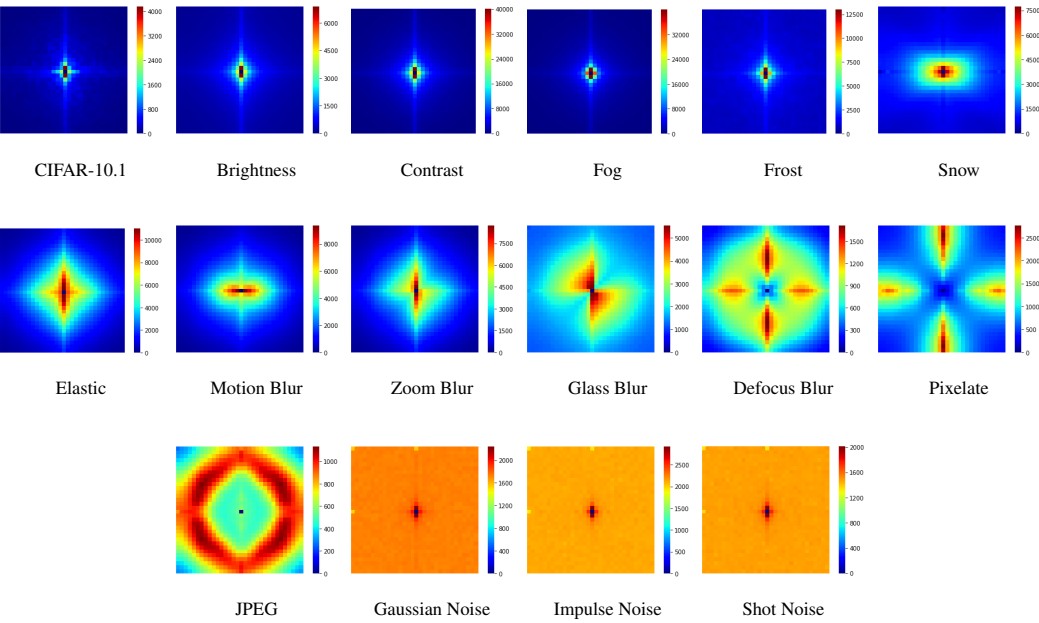

Figure 7: Power spectral densities for low (CIFAR-10.1, brightness, contrast, fog, frost, snow), mid (elastic transform, motion blur, zoom blur, glass blur, defocus blur, pixelate, JPEG compression), and high (Gaussian noise, impulse noise, shot noise) frequency shifts with respect to CIFAR-10.

## A.6 Pruning results on additional corruptions from CIFAR-10-C

Figure 8 and Table 5 extend Figure 2 and Table 1 to the nine remaining corruptions from CIFAR-10-C.

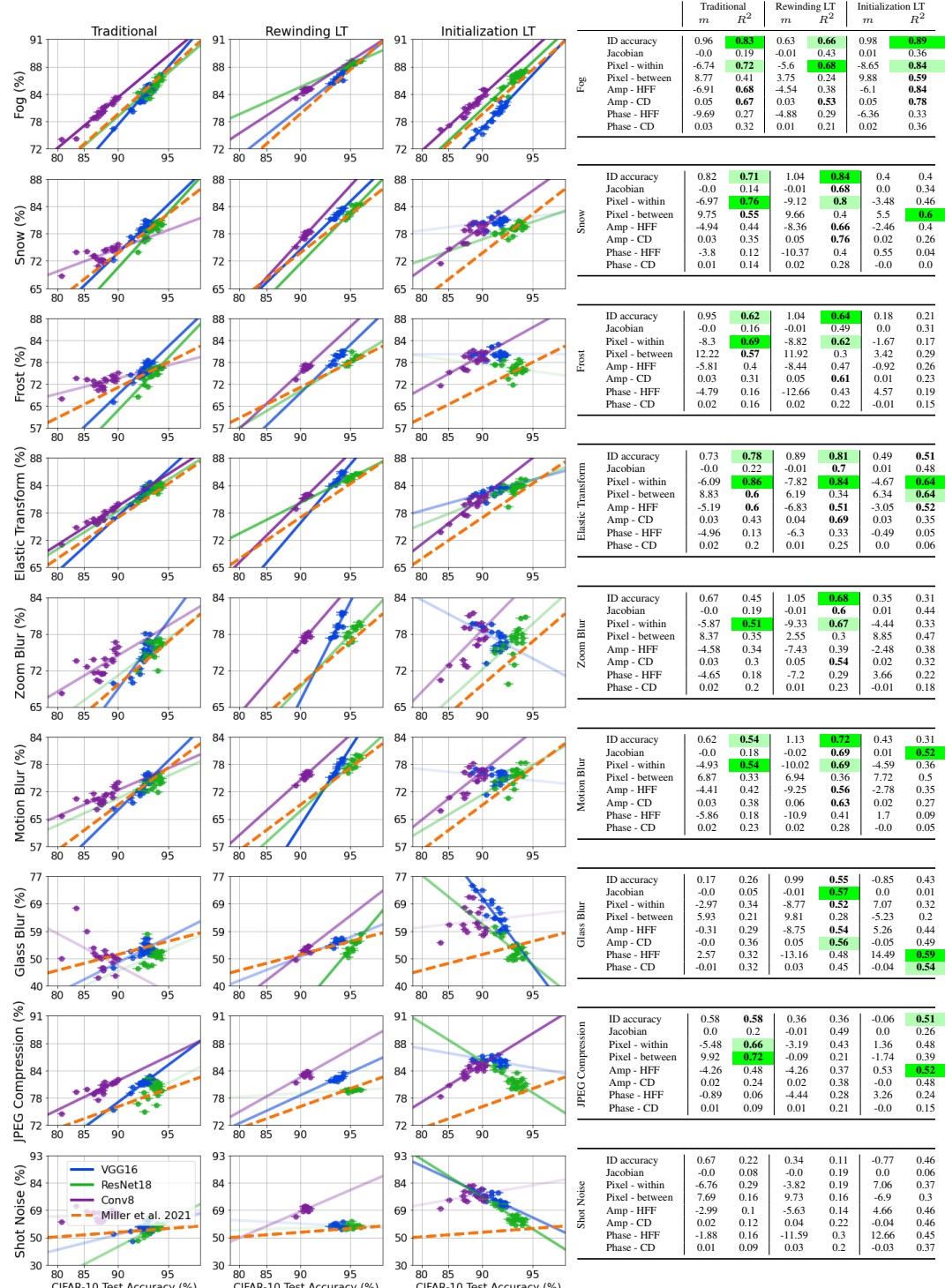

Figure 8: When does pruning affect OOD robustness?

Table 5: How well do metrics for pruned models correlate with OOD robustness?

## A.7 Augmentation results on additional corruptions from CIFAR-10-C

Figure 9 and Table 6 extend Figure 3 and Table 2 to the nine remaining corruptions from CIFAR-10-C.

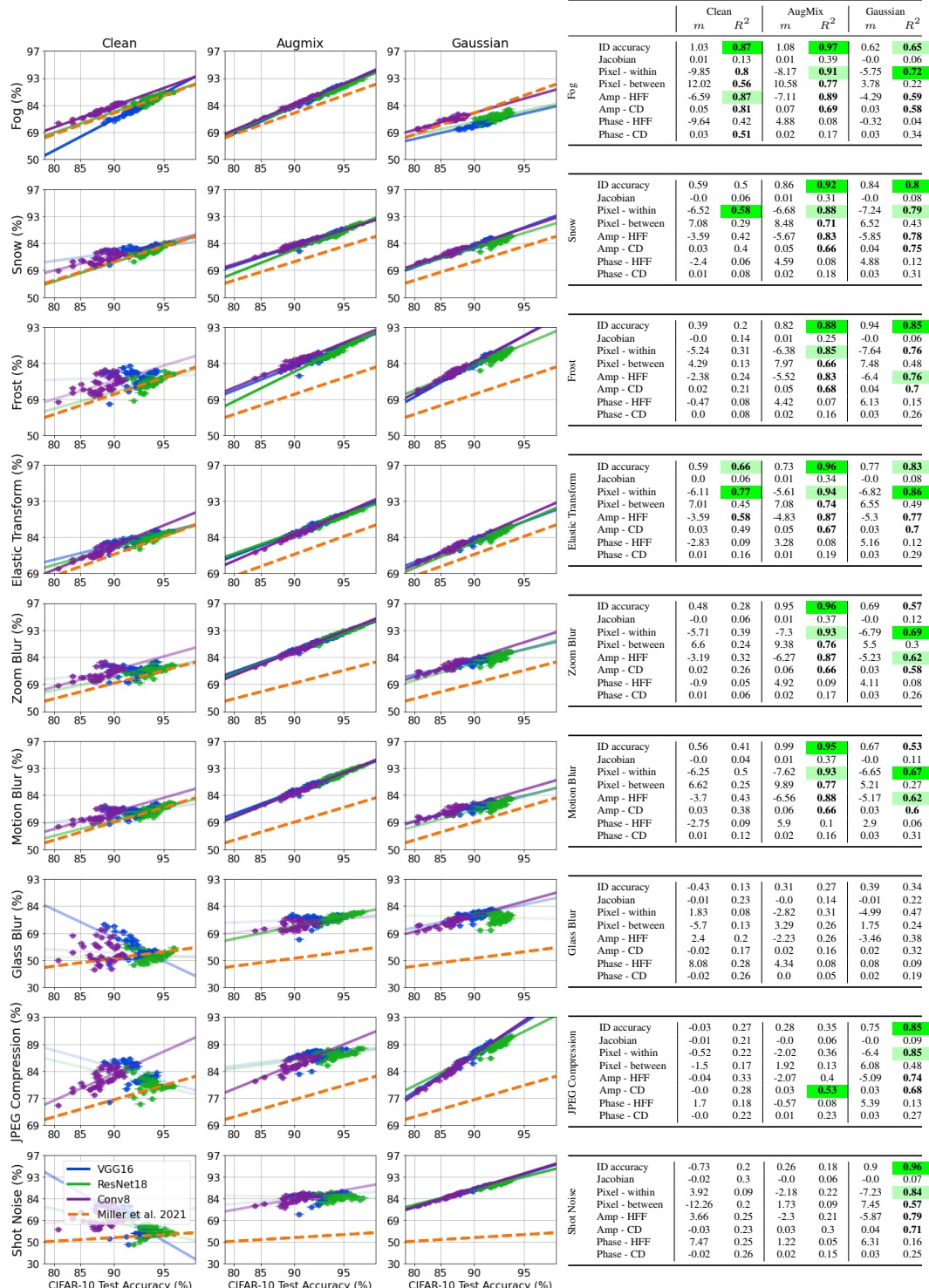

Figure 9: When does augmentation affect OOD robustness?

Table 6: How well do metrics for data-augmented models correlate with OOD robustness?

| | | Clean $m$ | Clean $R^2$ | AugMix $m$ | AugMix $R^2$ | Gaussian $m$ | Gaussian $R^2$ |
|---|---|---|---|---|---|---|---|
| **Fog** | ID accuracy | 1.03 | **0.87** | 1.08 | **0.97** | 0.62 | **0.65** |
| | Jacobian | 0.01 | 0.13 | 0.01 | 0.39 | -0.0 | 0.06 |
| | Pixel - within | -9.85 | 0.8 | -8.17 | **0.91** | -5.75 | **0.72** |
| | Pixel - between | 12.02 | 0.56 | 10.58 | **0.77** | 3.78 | 0.22 |
| | Amp - HFF | -6.59 | **0.87** | -7.11 | **0.89** | -4.29 | **0.59** |
| | Amp - CD | 0.05 | **0.81** | 0.07 | **0.69** | 0.03 | **0.58** |
| | Phase - HFF | -9.64 | 0.42 | 4.88 | 0.08 | -0.32 | 0.04 |
| | Phase - CD | 0.03 | **0.51** | 0.02 | 0.17 | 0.03 | 0.34 |
| **Snow** | ID accuracy | 0.59 | 0.5 | 0.86 | **0.92** | 0.84 | **0.8** |
| | Jacobian | -0.0 | 0.06 | 0.01 | 0.31 | -0.0 | 0.08 |
| | Pixel - within | -6.52 | **0.58** | -6.68 | **0.88** | -7.24 | **0.79** |
| | Pixel - between | 7.08 | 0.29 | 8.48 | **0.71** | 6.52 | 0.43 |
| | Amp - HFF | -3.59 | 0.42 | -5.67 | **0.83** | -5.85 | **0.78** |
| | Amp - CD | 0.03 | 0.4 | 0.05 | **0.66** | 0.04 | **0.75** |
| | Phase - HFF | -2.4 | 0.06 | 4.59 | 0.08 | 4.88 | 0.12 |
| | Phase - CD | 0.01 | 0.08 | 0.02 | 0.18 | 0.03 | 0.31 |
| **Frost** | ID accuracy | 0.39 | 0.2 | 0.82 | **0.88** | 0.94 | **0.85** |
| | Jacobian | -0.0 | 0.14 | 0.01 | 0.25 | -0.0 | 0.06 |
| | Pixel - within | -5.24 | 0.31 | -6.38 | **0.85** | -7.64 | **0.76** |
| | Pixel - between | 4.29 | 0.13 | 7.97 | **0.66** | 7.48 | 0.48 |
| | Amp - HFF | -2.38 | 0.24 | -5.52 | **0.83** | -6.4 | **0.76** |
| | Amp - CD | 0.02 | 0.21 | 0.05 | **0.68** | 0.04 | **0.7** |
| | Phase - HFF | -0.47 | 0.08 | 4.42 | 0.07 | 6.13 | 0.15 |
| | Phase - CD | 0.0 | 0.08 | 0.02 | 0.16 | 0.03 | 0.26 |
| **Elastic Transform** | ID accuracy | 0.59 | **0.66** | 0.73 | **0.96** | 0.77 | **0.83** |
| | Jacobian | 0.0 | 0.06 | 0.01 | 0.34 | -0.0 | 0.08 |
| | Pixel - within | -6.11 | **0.77** | -5.61 | **0.94** | -6.82 | **0.86** |
| | Pixel - between | 7.01 | 0.45 | 7.08 | **0.74** | 6.55 | 0.49 |
| | Amp - HFF | -3.59 | **0.58** | -4.83 | **0.87** | -5.3 | **0.77** |
| | Amp - CD | 0.03 | 0.49 | 0.05 | **0.67** | 0.03 | **0.7** |
| | Phase - HFF | -2.83 | 0.09 | 3.28 | 0.08 | 5.16 | 0.12 |
| | Phase - CD | 0.01 | 0.16 | 0.01 | 0.19 | 0.03 | 0.29 |
| **Zoom Blur** | ID accuracy | 0.48 | 0.28 | 0.95 | **0.96** | 0.69 | **0.57** |
| | Jacobian | -0.0 | 0.06 | 0.01 | 0.37 | -0.0 | 0.12 |
| | Pixel - within | -5.71 | 0.39 | -7.3 | **0.93** | -6.79 | **0.69** |
| | Pixel - between | 6.6 | 0.24 | 9.38 | **0.76** | 5.5 | 0.3 |
| | Amp - HFF | -3.19 | 0.32 | -6.27 | **0.87** | -5.23 | **0.62** |
| | Amp - CD | 0.02 | 0.26 | 0.06 | **0.66** | 0.03 | **0.58** |
| | Phase - HFF | -0.9 | 0.05 | 4.92 | 0.09 | 4.11 | 0.08 |
| | Phase - CD | 0.01 | 0.06 | 0.02 | 0.17 | 0.03 | 0.26 |
| **Motion Blur** | ID accuracy | 0.56 | 0.41 | 0.99 | **0.95** | 0.67 | **0.53** |
| | Jacobian | -0.0 | 0.04 | 0.01 | 0.37 | -0.0 | 0.11 |
| | Pixel - within | -6.25 | 0.5 | -7.62 | **0.93** | -6.65 | **0.67** |
| | Pixel - between | 6.62 | 0.25 | 9.89 | **0.77** | 5.21 | 0.27 |
| | Amp - HFF | -3.7 | 0.43 | -6.56 | **0.88** | -5.17 | **0.62** |
| | Amp - CD | 0.03 | 0.38 | 0.06 | **0.66** | 0.03 | **0.6** |
| | Phase - HFF | -2.75 | 0.09 | 5.9 | 0.1 | 2.9 | 0.06 |
| | Phase - CD | 0.01 | 0.12 | 0.02 | 0.16 | 0.03 | 0.31 |
| **Glass Blur** | ID accuracy | -0.43 | 0.13 | 0.31 | 0.27 | 0.39 | 0.34 |
| | Jacobian | -0.01 | 0.23 | -0.0 | 0.14 | -0.01 | 0.22 |
| | Pixel - within | 1.83 | 0.08 | -2.82 | 0.31 | -4.99 | 0.47 |
| | Pixel - between | -5.7 | 0.13 | 3.29 | 0.26 | 1.75 | 0.24 |
| | Amp - HFF | 2.4 | 0.2 | -2.23 | 0.26 | -3.46 | 0.38 |
| | Amp - CD | -0.02 | 0.17 | 0.02 | 0.16 | 0.02 | 0.32 |
| | Phase - HFF | 8.08 | 0.28 | 4.34 | 0.08 | 0.08 | 0.09 |
| | Phase - CD | -0.02 | 0.26 | 0.0 | 0.05 | 0.02 | 0.19 |
| **JPEG Compression** | ID accuracy | -0.03 | 0.27 | 0.28 | 0.35 | 0.75 | **0.85** |
| | Jacobian | -0.01 | 0.21 | -0.0 | 0.06 | -0.0 | 0.09 |
| | Pixel - within | -0.52 | 0.22 | -2.02 | 0.36 | -6.4 | **0.85** |
| | Pixel - between | -1.5 | 0.17 | 1.92 | 0.13 | 6.08 | 0.48 |
| | Amp - HFF | -0.04 | 0.33 | -2.07 | 0.4 | -5.09 | **0.74** |
| | Amp - CD | -0.0 | 0.28 | 0.03 | **0.53** | 0.03 | 0.68 |
| | Phase - HFF | 1.7 | 0.18 | -0.57 | 0.08 | 5.39 | 0.13 |
| | Phase - CD | -0.0 | 0.22 | 0.01 | 0.23 | 0.03 | 0.27 |
| **Shot Noise** | ID accuracy | -0.73 | 0.2 | 0.26 | 0.18 | 0.9 | **0.96** |
| | Jacobian | -0.02 | 0.3 | -0.0 | 0.06 | -0.0 | 0.07 |
| | Pixel - within | 3.92 | 0.09 | -2.18 | 0.22 | -7.23 | **0.84** |
| | Pixel - between | -12.26 | 0.2 | 1.73 | 0.09 | 7.45 | 0.57 |
| | Amp - HFF | 3.66 | 0.25 | -2.3 | 0.21 | -5.87 | **0.79** |
| | Amp - CD | -0.03 | 0.23 | 0.03 | 0.3 | 0.04 | **0.71** |
| | Phase - HFF | 7.47 | 0.25 | 1.22 | 0.05 | 6.31 | 0.16 |
| | Phase - CD | -0.02 | 0.26 | 0.02 | 0.15 | 0.03 | 0.25 |

## A.8 Best metric performance visualized for CIFAR-10 distribution shifts

Figures 10 and 11 visualize the fit to the data associated with the best metrics from the CIFAR-10 pruning analysis figures (Figures 2 and 8) and data augmentation analysis figures (Figures 3 and 9),

respectively. Note that Tables 7 and 8 are duplicates of the original figures' associated tables and are reproduced here for convenience.

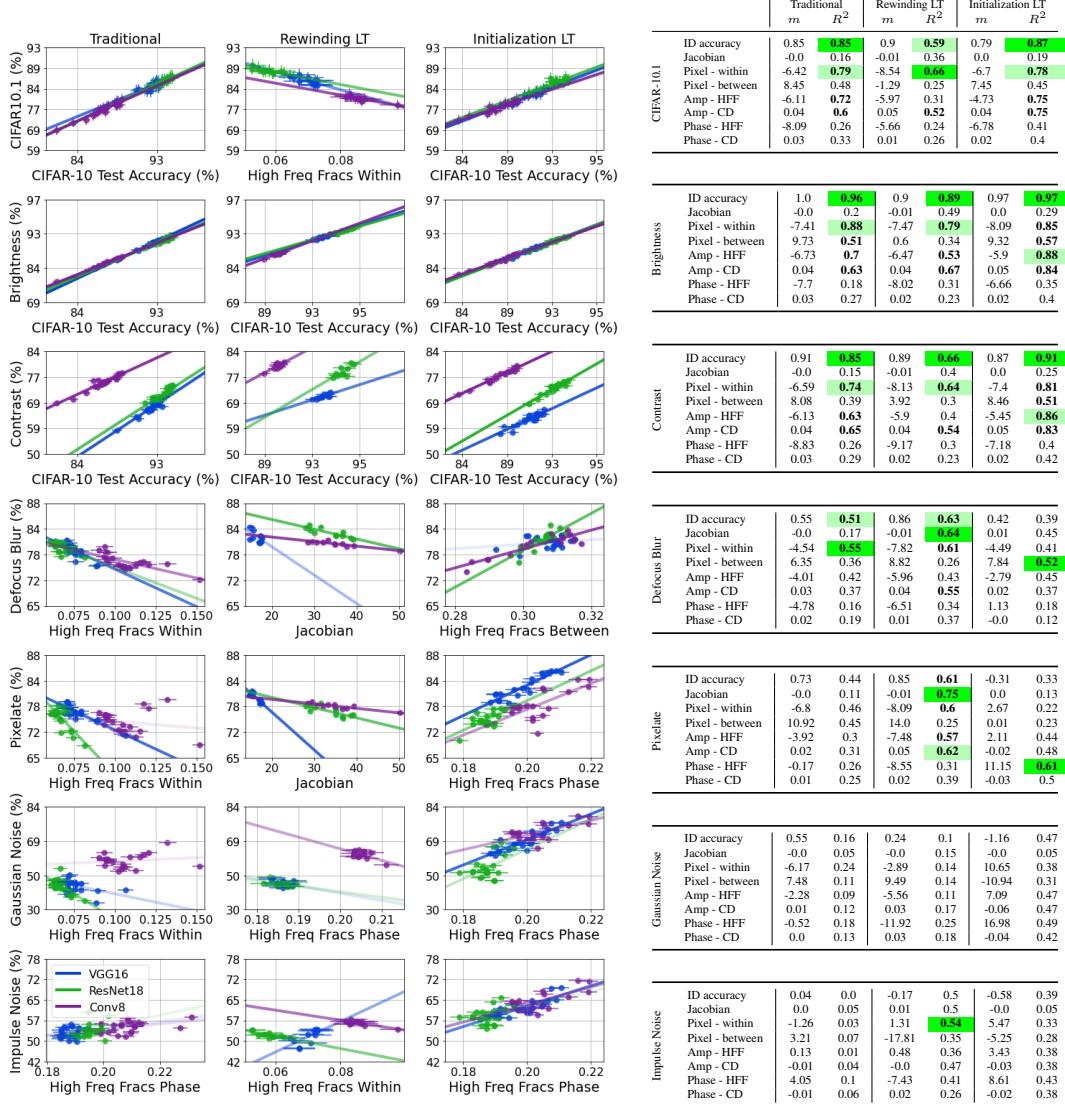

**CIFAR-10.1**

| | Traditional m | Traditional $R^2$ | Rewinding LT m | Rewinding LT $R^2$ | Initialization LT m | Initialization LT $R^2$ |
|---|---|---|---|---|---|---|
| ID accuracy | 0.85 | **0.85** | 0.9 | **0.59** | 0.79 | **0.87** |
| Jacobian | -0.0 | 0.16 | -0.01 | 0.36 | 0.0 | 0.19 |
| Pixel - within | -6.42 | **0.79** | -8.54 | **0.66** | -6.7 | **0.78** |
| Pixel - between | 8.45 | 0.48 | -1.29 | 0.25 | 7.45 | 0.45 |
| Amp - HFF | -6.11 | **0.72** | -5.97 | 0.31 | -4.73 | **0.75** |
| Amp - CD | 0.04 | **0.6** | 0.05 | **0.52** | 0.04 | **0.75** |
| Phase - HFF | -8.09 | 0.26 | -5.66 | 0.24 | -6.78 | 0.41 |
| Phase - CD | 0.03 | 0.33 | 0.01 | 0.26 | 0.02 | 0.4 |

**Brightness**

| | Traditional m | Traditional $R^2$ | Rewinding LT m | Rewinding LT $R^2$ | Initialization LT m | Initialization LT $R^2$ |
|---|---|---|---|---|---|---|
| ID accuracy | 1.0 | **0.96** | 0.9 | **0.89** | 0.97 | **0.97** |
| Jacobian | -0.0 | 0.2 | -0.01 | 0.49 | 0.0 | 0.29 |
| Pixel - within | -7.41 | **0.88** | -7.47 | **0.79** | -8.09 | **0.85** |
| Pixel - between | 9.73 | **0.51** | 0.6 | 0.34 | 9.32 | **0.57** |
| Amp - HFF | -6.73 | **0.7** | -6.47 | **0.53** | -5.9 | **0.88** |
| Amp - CD | 0.04 | **0.63** | 0.04 | **0.67** | 0.05 | **0.84** |
| Phase - HFF | -7.7 | 0.18 | -8.02 | 0.31 | -6.66 | 0.35 |
| Phase - CD | 0.03 | 0.27 | 0.02 | 0.23 | 0.02 | 0.4 |

**Contrast**

| | Traditional m | Traditional $R^2$ | Rewinding LT m | Rewinding LT $R^2$ | Initialization LT m | Initialization LT $R^2$ |
|---|---|---|---|---|---|---|
| ID accuracy | 0.91 | **0.85** | 0.89 | **0.66** | 0.87 | **0.91** |
| Jacobian | -0.0 | 0.15 | -0.01 | 0.4 | 0.0 | 0.25 |
| Pixel - within | -6.59 | **0.74** | -8.13 | **0.64** | -7.4 | **0.81** |
| Pixel - between | 8.08 | 0.39 | 3.92 | 0.3 | 8.46 | **0.51** |
| Amp - HFF | -6.13 | **0.63** | -5.9 | 0.4 | -5.45 | **0.86** |
| Amp - CD | 0.04 | **0.65** | 0.04 | **0.54** | 0.05 | **0.83** |
| Phase - HFF | -8.83 | 0.26 | -9.17 | 0.3 | -7.18 | 0.4 |
| Phase - CD | 0.03 | 0.29 | 0.02 | 0.23 | 0.02 | 0.42 |

**Defocus Blur**

| | Traditional m | Traditional $R^2$ | Rewinding LT m | Rewinding LT $R^2$ | Initialization LT m | Initialization LT $R^2$ |
|---|---|---|---|---|---|---|
| ID accuracy | 0.55 | **0.51** | 0.86 | **0.63** | 0.42 | 0.39 |
| Jacobian | -0.0 | 0.17 | -0.01 | **0.64** | 0.01 | 0.45 |
| Pixel - within | -4.54 | **0.55** | -7.82 | **0.61** | -4.49 | 0.41 |
| Pixel - between | 6.35 | 0.36 | 8.82 | 0.26 | 7.84 | **0.52** |
| Amp - HFF | -4.01 | 0.42 | -5.96 | 0.43 | -2.79 | 0.45 |
| Amp - CD | 0.03 | 0.37 | 0.04 | **0.55** | 0.02 | 0.37 |
| Phase - HFF | -4.78 | 0.16 | -6.51 | 0.34 | 1.13 | 0.18 |
| Phase - CD | 0.02 | 0.19 | 0.01 | 0.37 | -0.0 | 0.12 |

**Pixelate**

| | Traditional m | Traditional $R^2$ | Rewinding LT m | Rewinding LT $R^2$ | Initialization LT m | Initialization LT $R^2$ |
|---|---|---|---|---|---|---|
| ID accuracy | 0.73 | 0.44 | 0.85 | **0.61** | -0.31 | 0.33 |
| Jacobian | -0.0 | 0.11 | -0.01 | **0.75** | 0.0 | 0.13 |
| Pixel - within | -6.8 | 0.46 | -8.09 | **0.6** | 2.67 | 0.22 |
| Pixel - between | 10.92 | 0.45 | 14.0 | 0.25 | 0.01 | 0.23 |
| Amp - HFF | -3.92 | 0.3 | -7.48 | **0.57** | 2.11 | 0.44 |
| Amp - CD | 0.02 | 0.31 | 0.05 | **0.62** | -0.02 | 0.48 |
| Phase - HFF | -0.17 | 0.26 | -8.55 | 0.31 | 11.15 | **0.61** |
| Phase - CD | 0.01 | 0.25 | 0.02 | 0.39 | -0.03 | 0.5 |

**Gaussian Noise**

| | Traditional m | Traditional $R^2$ | Rewinding LT m | Rewinding LT $R^2$ | Initialization LT m | Initialization LT $R^2$ |
|---|---|---|---|---|---|---|
| ID accuracy | 0.55 | 0.16 | 0.24 | 0.1 | -1.16 | 0.47 |
| Jacobian | -0.0 | 0.05 | -0.0 | 0.15 | -0.0 | 0.05 |
| Pixel - within | -6.17 | 0.24 | -2.89 | 0.14 | 10.65 | 0.38 |
| Pixel - between | 7.48 | 0.11 | 9.49 | 0.14 | -10.94 | 0.31 |
| Amp - HFF | -2.28 | 0.09 | -5.56 | 0.11 | 7.09 | 0.47 |
| Amp - CD | 0.01 | 0.12 | 0.03 | 0.17 | -0.06 | 0.47 |
| Phase - HFF | -0.52 | 0.18 | -11.92 | 0.25 | 16.98 | 0.49 |
| Phase - CD | 0.0 | 0.13 | 0.03 | 0.18 | -0.04 | 0.42 |

**Impulse Noise**

| | Traditional m | Traditional $R^2$ | Rewinding LT m | Rewinding LT $R^2$ | Initialization LT m | Initialization LT $R^2$ |
|---|---|---|---|---|---|---|
| ID accuracy | 0.04 | 0.0 | -0.17 | 0.5 | -0.58 | 0.39 |
| Jacobian | 0.0 | 0.05 | 0.01 | 0.5 | -0.0 | 0.05 |
| Pixel - within | -1.26 | 0.03 | 1.31 | **0.54** | 5.47 | 0.33 |
| Pixel - between | 3.21 | 0.07 | -17.81 | 0.35 | -5.25 | 0.28 |
| Amp - HFF | 0.13 | 0.01 | 0.48 | 0.36 | 3.43 | 0.38 |
| Amp - CD | -0.01 | 0.04 | -0.0 | 0.47 | -0.03 | 0.38 |
| Phase - HFF | 4.05 | 0.1 | -7.43 | 0.41 | 8.61 | 0.43 |
| Phase - CD | -0.01 | 0.06 | 0.02 | 0.26 | -0.02 | 0.38 |

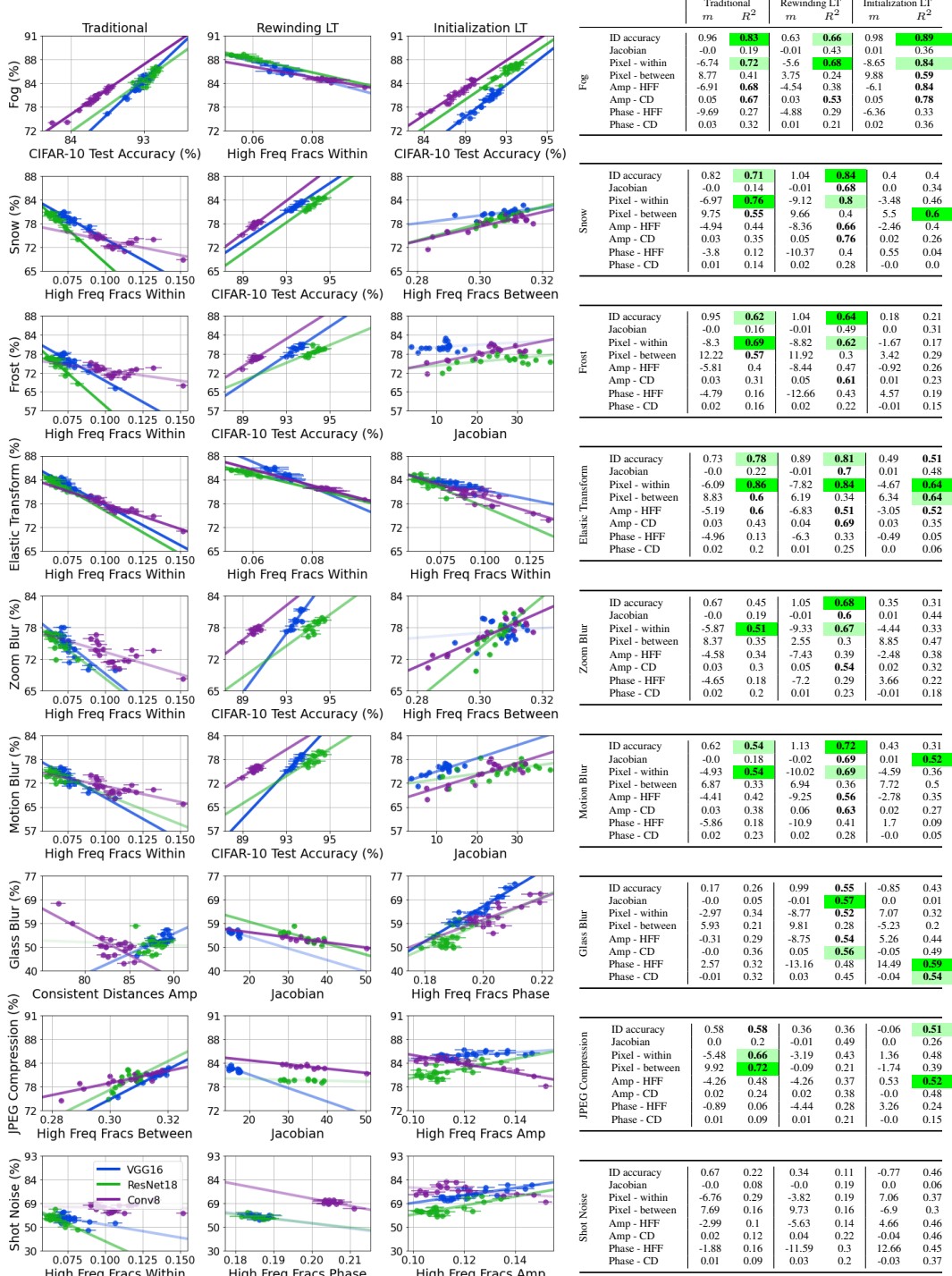

Figure 10: When does pruning affect OOD robustness?

| | | Traditional | | Rewinding LT | | Initialization LT | |
|---|---|---|---|---|---|---|---|
| | | $m$ | $R^2$ | $m$ | $R^2$ | $m$ | $R^2$ |
| Fog | ID accuracy | 0.96 | **0.83** | 0.63 | **0.66** | 0.98 | **0.89** |
| | Jacobian | -0.0 | 0.19 | -0.01 | 0.43 | 0.01 | 0.36 |
| | Pixel - within | -6.74 | **0.72** | -5.6 | **0.68** | -8.65 | **0.84** |
| | Pixel - between | 8.77 | 0.41 | 3.75 | 0.24 | 9.88 | **0.59** |
| | Amp - HFF | -6.91 | **0.68** | -4.54 | 0.38 | -6.1 | **0.84** |
| | Amp - CD | 0.05 | **0.67** | 0.03 | **0.53** | 0.05 | **0.78** |
| | Phase - HFF | -9.69 | 0.27 | -4.88 | 0.29 | -6.36 | 0.33 |
| | Phase - CD | 0.03 | 0.32 | 0.01 | 0.21 | 0.02 | 0.36 |
| Snow | ID accuracy | 0.82 | **0.71** | 1.04 | **0.84** | 0.4 | 0.4 |
| | Jacobian | -0.0 | 0.14 | -0.01 | **0.68** | 0.0 | 0.34 |
| | Pixel - within | -6.97 | **0.76** | -9.12 | **0.8** | -3.48 | 0.46 |
| | Pixel - between | 9.75 | **0.55** | 9.66 | 0.4 | 5.5 | **0.6** |
| | Amp - HFF | -4.94 | 0.44 | -8.36 | **0.66** | -2.46 | 0.4 |
| | Amp - CD | 0.03 | 0.35 | 0.05 | **0.76** | 0.02 | 0.26 |
| | Phase - HFF | -3.8 | 0.12 | -10.37 | 0.4 | 0.55 | 0.04 |
| | Phase - CD | 0.01 | 0.14 | 0.02 | 0.28 | -0.0 | 0.0 |
| Frost | ID accuracy | 0.95 | **0.62** | 1.04 | **0.64** | 0.18 | 0.21 |
| | Jacobian | -0.0 | 0.16 | -0.01 | 0.49 | 0.0 | 0.31 |
| | Pixel - within | -8.3 | **0.69** | -8.82 | **0.62** | -1.67 | 0.17 |
| | Pixel - between | 12.22 | **0.57** | 11.92 | 0.3 | 3.42 | 0.29 |
| | Amp - HFF | -5.81 | 0.4 | -8.44 | 0.47 | -0.92 | 0.26 |
| | Amp - CD | 0.03 | 0.31 | 0.05 | **0.61** | 0.01 | 0.23 |
| | Phase - HFF | -4.79 | 0.16 | -12.66 | 0.43 | 4.57 | 0.19 |
| | Phase - CD | 0.02 | 0.16 | 0.02 | 0.22 | -0.01 | 0.15 |
| Elastic Transform | ID accuracy | 0.73 | **0.78** | 0.89 | **0.81** | 0.49 | **0.51** |
| | Jacobian | -0.0 | 0.22 | -0.01 | **0.7** | 0.01 | 0.48 |
| | Pixel - within | -6.09 | **0.86** | -7.82 | **0.84** | -4.67 | **0.64** |
| | Pixel - between | 8.83 | **0.6** | 6.19 | 0.34 | 6.34 | **0.64** |
| | Amp - HFF | -5.19 | **0.6** | -6.83 | **0.51** | -3.05 | **0.52** |
| | Amp - CD | 0.03 | 0.43 | 0.04 | **0.69** | 0.03 | 0.35 |
| | Phase - HFF | -4.96 | 0.13 | -6.3 | 0.33 | -0.49 | 0.05 |
| | Phase - CD | 0.02 | 0.2 | 0.01 | 0.25 | 0.0 | 0.06 |
| Zoom Blur | ID accuracy | 0.67 | 0.45 | 1.05 | **0.68** | 0.35 | 0.31 |
| | Jacobian | -0.0 | 0.19 | -0.01 | **0.6** | 0.01 | 0.44 |
| | Pixel - within | -5.87 | **0.51** | -9.33 | **0.67** | -4.44 | 0.33 |
| | Pixel - between | 8.37 | 0.35 | 2.55 | 0.3 | 8.85 | 0.47 |
| | Amp - HFF | -4.58 | 0.34 | -7.43 | 0.39 | -2.48 | 0.38 |
| | Amp - CD | 0.03 | 0.3 | 0.05 | **0.54** | 0.02 | 0.32 |
| | Phase - HFF | -4.65 | 0.18 | -7.2 | 0.29 | 3.66 | 0.22 |
| | Phase - CD | 0.02 | 0.2 | 0.01 | 0.23 | -0.01 | 0.18 |
| Motion Blur | ID accuracy | 0.62 | **0.54** | 1.13 | **0.72** | 0.43 | 0.31 |
| | Jacobian | -0.0 | 0.18 | -0.02 | **0.69** | 0.01 | **0.52** |
| | Pixel - within | -4.93 | **0.54** | -10.02 | **0.69** | -4.59 | 0.36 |
| | Pixel - between | 6.87 | 0.33 | 6.94 | 0.36 | 7.72 | 0.5 |
| | Amp - HFF | -4.41 | 0.42 | -9.25 | **0.56** | -2.78 | 0.35 |
| | Amp - CD | 0.03 | 0.38 | 0.06 | **0.63** | 0.02 | 0.27 |
| | Phase - HFF | -5.86 | 0.18 | -10.9 | 0.41 | 1.7 | 0.09 |
| | Phase - CD | 0.02 | 0.23 | 0.02 | 0.28 | -0.0 | 0.05 |
| Glass Blur | ID accuracy | 0.17 | 0.26 | 0.99 | **0.55** | -0.85 | 0.43 |
| | Jacobian | -0.0 | 0.05 | -0.01 | **0.57** | 0.0 | 0.01 |
| | Pixel - within | -2.97 | 0.34 | -8.77 | **0.52** | 7.07 | 0.32 |
| | Pixel - between | 5.93 | 0.21 | 9.81 | 0.28 | -5.23 | 0.2 |
| | Amp - HFF | -0.31 | 0.29 | -8.75 | **0.54** | 5.26 | 0.44 |
| | Amp - CD | -0.0 | 0.36 | 0.05 | **0.56** | -0.05 | 0.49 |
| | Phase - HFF | 2.57 | 0.32 | -13.16 | 0.48 | 14.49 | **0.59** |
| | Phase - CD | -0.01 | 0.32 | 0.03 | 0.45 | -0.04 | **0.54** |
| JPEG Compression | ID accuracy | 0.58 | **0.58** | 0.36 | 0.36 | -0.06 | **0.51** |
| | Jacobian | 0.0 | 0.2 | -0.01 | 0.49 | 0.0 | 0.26 |
| | Pixel - within | -5.48 | **0.66** | -3.19 | 0.43 | 1.36 | 0.48 |
| | Pixel - between | 9.92 | **0.72** | -0.09 | 0.21 | -1.74 | 0.39 |
| | Amp - HFF | -4.26 | 0.48 | -4.26 | 0.37 | 0.53 | **0.52** |
| | Amp - CD | 0.02 | 0.24 | 0.02 | 0.38 | -0.0 | 0.48 |
| | Phase - HFF | -0.89 | 0.06 | -4.44 | 0.28 | 3.26 | 0.24 |
| | Phase - CD | 0.01 | 0.09 | 0.01 | 0.21 | -0.0 | 0.15 |
| Shot Noise | ID accuracy | 0.67 | 0.22 | 0.34 | 0.11 | -0.77 | 0.46 |
| | Jacobian | -0.0 | 0.08 | -0.0 | 0.19 | 0.0 | 0.06 |
| | Pixel - within | -6.76 | 0.29 | -3.82 | 0.19 | 7.06 | 0.37 |
| | Pixel - between | 7.69 | 0.16 | 9.73 | 0.16 | -6.9 | 0.3 |
| | Amp - HFF | -2.99 | 0.1 | -5.63 | 0.14 | 4.66 | 0.46 |
| | Amp - CD | 0.02 | 0.12 | 0.04 | 0.22 | -0.04 | 0.46 |
| | Phase - HFF | -1.88 | 0.16 | -11.59 | 0.3 | 12.66 | 0.45 |
| | Phase - CD | 0.01 | 0.09 | 0.03 | 0.2 | -0.03 | 0.37 |

Table 7: Can model metrics explain why pruning affects OOD robustness?

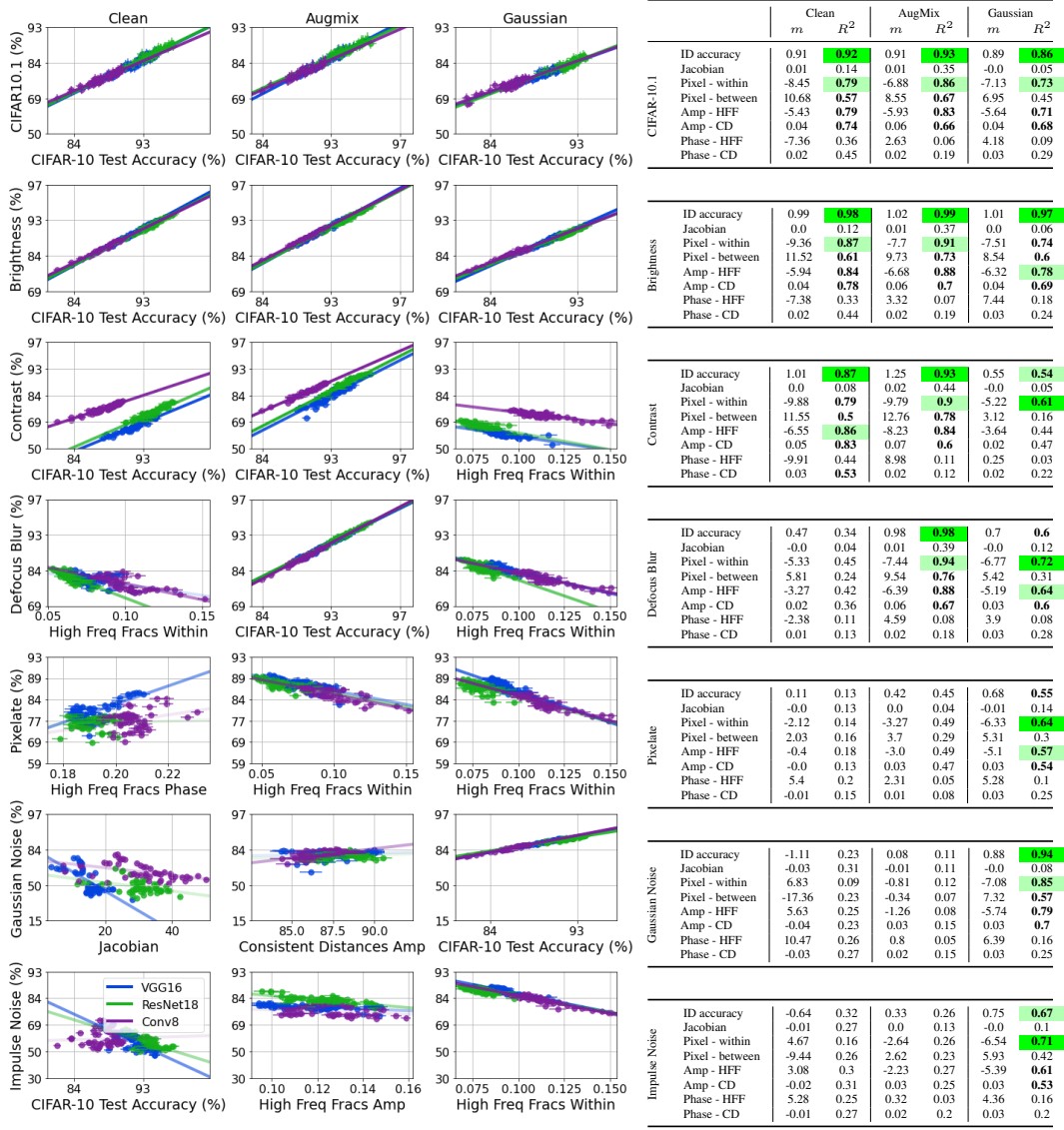

| | Clean | | AugMix | | Gaussian | |
|---|---|---|---|---|---|---|
| | $m$ | $R^2$ | $m$ | $R^2$ | $m$ | $R^2$ |
| **CIFAR-10.1** | | | | | | |
| ID accuracy | 0.91 | 0.92 | 0.91 | 0.93 | 0.89 | 0.86 |
| Jacobian | 0.01 | 0.14 | 0.01 | 0.35 | 0.0 | 0.05 |
| Pixel - within | -8.45 | 0.79 | -6.88 | 0.86 | -7.13 | 0.73 |
| Pixel - between | 10.68 | 0.57 | 8.55 | 0.67 | 6.95 | 0.45 |
| Amp - HFF | -5.43 | 0.79 | -5.93 | 0.83 | -5.64 | 0.71 |
| Amp - CD | 0.04 | 0.74 | 0.06 | 0.66 | 0.04 | 0.68 |
| Phase - HFF | -7.36 | 0.36 | 2.63 | 0.06 | 4.18 | 0.09 |
| Phase - CD | 0.02 | 0.45 | 0.02 | 0.19 | 0.03 | 0.29 |
| **Brightness** | | | | | | |
| ID accuracy | 0.99 | 0.98 | 1.02 | 0.99 | 1.01 | 0.97 |
| Jacobian | 0.0 | 0.12 | 0.01 | 0.37 | 0.0 | 0.06 |
| Pixel - within | -9.36 | 0.87 | -7.7 | 0.91 | -7.51 | 0.74 |
| Pixel - between | 11.52 | 0.61 | 9.73 | 0.73 | 8.54 | 0.6 |
| Amp - HFF | -5.94 | 0.84 | -6.68 | 0.88 | -6.32 | 0.78 |
| Amp - CD | 0.04 | 0.78 | 0.06 | 0.7 | 0.04 | 0.69 |
| Phase - HFF | -7.38 | 0.33 | 3.32 | 0.07 | 7.44 | 0.18 |
| Phase - CD | 0.02 | 0.44 | 0.02 | 0.19 | 0.03 | 0.24 |
| **Contrast** | | | | | | |
| ID accuracy | 1.01 | 0.87 | 1.25 | 0.93 | 0.55 | 0.54 |
| Jacobian | 0.0 | 0.08 | 0.02 | 0.44 | -0.0 | 0.05 |
| Pixel - within | -9.88 | 0.79 | -9.79 | 0.9 | -5.22 | 0.61 |
| Pixel - between | 11.55 | 0.5 | 12.76 | 0.78 | 3.12 | 0.16 |
| Amp - HFF | -6.55 | 0.86 | -8.23 | 0.84 | -3.64 | 0.44 |
| Amp - CD | 0.05 | 0.83 | 0.07 | 0.6 | 0.02 | 0.47 |
| Phase - HFF | -9.91 | 0.44 | 8.98 | 0.11 | 0.25 | 0.03 |
| Phase - CD | 0.03 | 0.53 | 0.02 | 0.12 | 0.02 | 0.22 |
| **Defocus Blur** | | | | | | |
| ID accuracy | 0.47 | 0.34 | 0.98 | 0.98 | 0.7 | 0.6 |
| Jacobian | -0.0 | 0.04 | 0.01 | 0.39 | -0.0 | 0.12 |
| Pixel - within | -5.33 | 0.45 | -7.44 | 0.94 | -6.77 | 0.72 |
| Pixel - between | 5.81 | 0.24 | 9.54 | 0.76 | 5.42 | 0.31 |
| Amp - HFF | -3.27 | 0.42 | -6.39 | 0.88 | -5.19 | 0.64 |
| Amp - CD | 0.02 | 0.36 | 0.06 | 0.67 | 0.03 | 0.6 |
| Phase - HFF | -2.38 | 0.11 | 4.59 | 0.08 | 3.9 | 0.08 |
| Phase - CD | 0.01 | 0.13 | 0.02 | 0.18 | 0.03 | 0.28 |
| **Pixelate** | | | | | | |
| ID accuracy | 0.11 | 0.13 | 0.42 | 0.45 | 0.68 | 0.55 |
| Jacobian | -0.0 | 0.13 | 0.0 | 0.04 | -0.01 | 0.14 |
| Pixel - within | -2.12 | 0.14 | -3.27 | 0.49 | -6.33 | 0.64 |
| Pixel - between | 2.03 | 0.16 | 3.7 | 0.29 | 5.31 | 0.3 |
| Amp - HFF | -0.4 | 0.18 | -3.0 | 0.49 | -5.1 | 0.57 |
| Amp - CD | -0.0 | 0.13 | 0.03 | 0.47 | 0.03 | 0.54 |
| Phase - HFF | 5.4 | 0.2 | 2.31 | 0.05 | 5.28 | 0.1 |
| Phase - CD | -0.01 | 0.15 | 0.01 | 0.08 | 0.03 | 0.25 |
| **Gaussian Noise** | | | | | | |
| ID accuracy | -1.11 | 0.23 | 0.08 | 0.11 | 0.88 | 0.94 |
| Jacobian | -0.03 | 0.31 | -0.01 | 0.11 | -0.0 | 0.08 |
| Pixel - within | 6.83 | 0.09 | -0.81 | 0.12 | -7.08 | 0.85 |
| Pixel - between | -17.36 | 0.23 | -0.34 | 0.07 | 7.32 | 0.57 |
| Amp - HFF | 5.63 | 0.25 | -1.26 | 0.08 | -5.74 | 0.79 |
| Amp - CD | -0.04 | 0.23 | 0.03 | 0.15 | 0.03 | 0.7 |
| Phase - HFF | 10.47 | 0.26 | 0.8 | 0.05 | 6.39 | 0.16 |
| Phase - CD | -0.03 | 0.27 | 0.02 | 0.15 | 0.03 | 0.25 |
| **Impulse Noise** | | | | | | |
| ID accuracy | -0.64 | 0.32 | 0.33 | 0.26 | 0.75 | 0.67 |
| Jacobian | -0.01 | 0.27 | 0.0 | 0.13 | -0.0 | 0.1 |
| Pixel - within | 4.67 | 0.16 | -2.64 | 0.26 | -6.54 | 0.71 |
| Pixel - between | -9.44 | 0.26 | 2.62 | 0.23 | 5.93 | 0.42 |
| Amp - HFF | 3.08 | 0.3 | -2.23 | 0.27 | -5.39 | 0.61 |
| Amp - CD | -0.02 | 0.31 | 0.03 | 0.25 | 0.03 | 0.53 |
| Phase - HFF | 5.28 | 0.25 | 0.32 | 0.03 | 4.36 | 0.16 |
| Phase - CD | -0.01 | 0.27 | 0.02 | 0.2 | 0.03 | 0.2 |

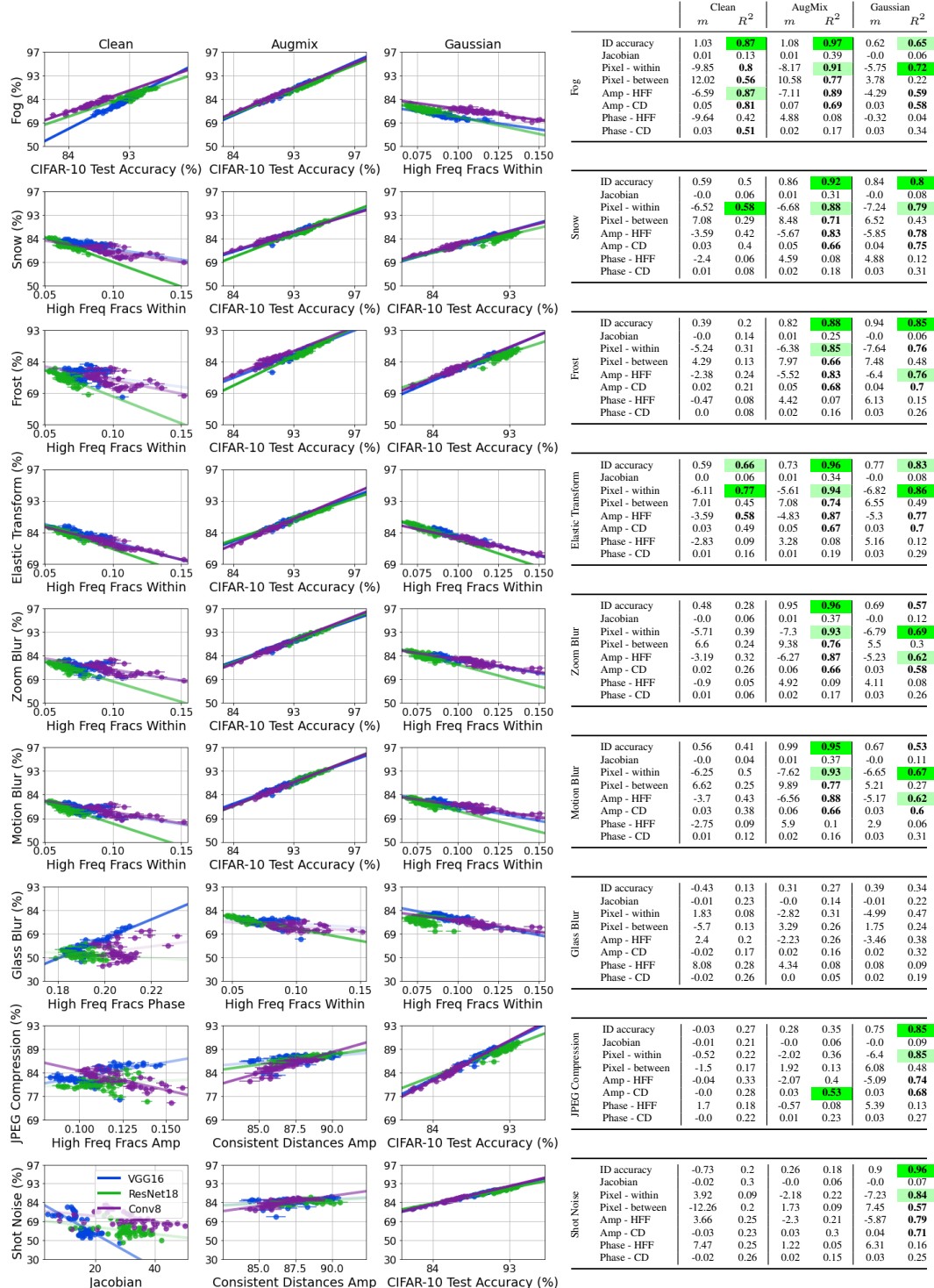

Figure 11: When does augmentation affect OOD robustness?

Table 8: Can model metrics explain why augmentation affects OOD robustness?

| | | Clean | | AugMix | | Gaussian | |
|---|---|---|---|---|---|---|---|
| | | $m$ | $R^2$ | $m$ | $R^2$ | $m$ | $R^2$ |
| **Fog** | ID accuracy | 1.03 | **0.87** | 1.08 | **0.97** | 0.62 | **0.65** |
| | Jacobian | 0.01 | 0.13 | 0.01 | 0.39 | -0.0 | 0.06 |
| | Pixel - within | -9.85 | **0.8** | -8.17 | **0.91** | -5.75 | **0.72** |
| | Pixel - between | 12.02 | **0.56** | 10.58 | **0.77** | 3.78 | 0.22 |
| | Amp - HFF | -6.59 | **0.87** | -7.11 | **0.89** | -4.29 | **0.59** |
| | Amp - CD | 0.05 | **0.81** | 0.07 | **0.69** | 0.03 | **0.58** |
| | Phase - HFF | -9.64 | 0.42 | 4.88 | 0.08 | -0.32 | 0.04 |
| | Phase - CD | 0.03 | **0.51** | 0.02 | 0.17 | 0.03 | 0.34 |
| **Snow** | ID accuracy | 0.59 | 0.5 | 0.86 | **0.92** | 0.84 | **0.8** |
| | Jacobian | -0.0 | 0.06 | 0.01 | 0.31 | -0.0 | 0.08 |
| | Pixel - within | -6.52 | **0.58** | -6.68 | **0.88** | -7.24 | **0.79** |
| | Pixel - between | 7.08 | 0.29 | 8.48 | **0.71** | 6.52 | 0.43 |
| | Amp - HFF | -3.59 | 0.42 | -5.67 | **0.83** | -5.85 | **0.78** |
| | Amp - CD | 0.03 | 0.4 | 0.05 | **0.66** | 0.04 | **0.75** |
| | Phase - HFF | -2.4 | 0.06 | 4.59 | 0.08 | 4.88 | 0.12 |
| | Phase - CD | 0.01 | 0.08 | 0.02 | 0.18 | 0.03 | 0.31 |
| **Frost** | ID accuracy | 0.39 | 0.2 | 0.82 | **0.88** | 0.94 | **0.85** |
| | Jacobian | -0.0 | 0.14 | 0.01 | 0.25 | -0.0 | 0.06 |
| | Pixel - within | -5.24 | 0.31 | -6.38 | **0.85** | -7.64 | **0.76** |
| | Pixel - between | 4.29 | 0.13 | 7.97 | **0.66** | 7.48 | 0.48 |
| | Amp - HFF | -2.38 | 0.24 | -5.52 | **0.83** | -6.4 | **0.76** |
| | Amp - CD | 0.02 | 0.21 | 0.05 | **0.68** | 0.04 | **0.7** |
| | Phase - HFF | -0.47 | 0.08 | 4.42 | 0.07 | 6.13 | 0.15 |
| | Phase - CD | 0.0 | 0.08 | 0.02 | 0.16 | 0.03 | 0.26 |
| **Elastic Transform** | ID accuracy | 0.59 | **0.66** | 0.73 | **0.96** | 0.77 | **0.83** |
| | Jacobian | 0.0 | 0.06 | 0.01 | 0.34 | -0.0 | 0.08 |
| | Pixel - within | -6.11 | **0.77** | -5.61 | **0.94** | -6.82 | **0.86** |
| | Pixel - between | 7.01 | 0.45 | 7.08 | **0.74** | 6.55 | 0.49 |
| | Amp - HFF | -3.59 | **0.58** | -4.83 | **0.87** | -5.3 | **0.77** |
| | Amp - CD | 0.03 | 0.49 | 0.05 | **0.67** | 0.03 | **0.7** |
| | Phase - HFF | -2.83 | 0.09 | 3.28 | 0.08 | 5.16 | 0.12 |
| | Phase - CD | 0.01 | 0.16 | 0.01 | 0.19 | 0.03 | 0.29 |
| **Zoom Blur** | ID accuracy | 0.48 | 0.28 | 0.95 | **0.96** | 0.69 | **0.57** |
| | Jacobian | -0.0 | 0.06 | 0.01 | 0.37 | -0.0 | 0.12 |
| | Pixel - within | -5.71 | 0.39 | -7.3 | **0.93** | -6.79 | **0.69** |
| | Pixel - between | 6.6 | 0.24 | 9.38 | **0.76** | 5.5 | 0.3 |
| | Amp - HFF | -3.19 | 0.32 | -6.27 | **0.87** | -5.23 | **0.62** |
| | Amp - CD | 0.02 | 0.26 | 0.06 | **0.66** | 0.03 | **0.58** |
| | Phase - HFF | -0.9 | 0.05 | 4.92 | 0.09 | 4.11 | 0.08 |
| | Phase - CD | 0.01 | 0.06 | 0.02 | 0.17 | 0.03 | 0.26 |
| **Motion Blur** | ID accuracy | 0.56 | 0.41 | 0.99 | **0.95** | 0.67 | **0.53** |
| | Jacobian | -0.0 | 0.04 | 0.01 | 0.37 | -0.0 | 0.11 |
| | Pixel - within | -6.25 | 0.5 | -7.62 | **0.93** | -6.65 | **0.67** |
| | Pixel - between | 6.62 | 0.25 | 9.89 | **0.77** | 5.21 | 0.27 |
| | Amp - HFF | -3.7 | 0.43 | -6.56 | **0.88** | -5.17 | **0.62** |
| | Amp - CD | 0.03 | 0.38 | 0.06 | **0.66** | 0.03 | **0.6** |
| | Phase - HFF | -2.75 | 0.09 | 5.9 | 0.1 | 2.9 | 0.06 |
| | Phase - CD | 0.01 | 0.12 | 0.02 | 0.16 | 0.03 | 0.31 |
| **Glass Blur** | ID accuracy | -0.43 | 0.13 | 0.31 | 0.27 | 0.39 | 0.34 |
| | Jacobian | -0.01 | 0.23 | -0.0 | 0.14 | -0.01 | 0.22 |
| | Pixel - within | 1.83 | 0.08 | -2.82 | 0.31 | -4.99 | 0.47 |
| | Pixel - between | -5.7 | 0.13 | 3.29 | 0.26 | 1.75 | 0.24 |
| | Amp - HFF | 2.4 | 0.2 | -2.23 | 0.26 | -3.46 | 0.38 |
| | Amp - CD | -0.02 | 0.17 | 0.02 | 0.16 | 0.02 | 0.32 |
| | Phase - HFF | 8.08 | 0.28 | 4.34 | 0.08 | 0.08 | 0.09 |
| | Phase - CD | -0.02 | 0.26 | 0.0 | 0.05 | 0.02 | 0.19 |
| **JPEG Compression** | ID accuracy | -0.03 | 0.27 | 0.28 | 0.35 | 0.75 | **0.85** |
| | Jacobian | -0.01 | 0.21 | -0.0 | 0.06 | -0.0 | 0.09 |
| | Pixel - within | -0.52 | 0.22 | -2.02 | 0.36 | -6.4 | **0.85** |
| | Pixel - between | -1.5 | 0.17 | 1.92 | 0.13 | 6.08 | 0.48 |
| | Amp - HFF | -0.04 | 0.33 | -2.07 | 0.4 | -5.09 | **0.74** |
| | Amp - CD | -0.0 | 0.28 | 0.03 | **0.53** | 0.03 | **0.68** |
| | Phase - HFF | 1.7 | 0.18 | -0.57 | 0.08 | 5.39 | 0.13 |
| | Phase - CD | -0.0 | 0.22 | 0.01 | 0.23 | 0.03 | 0.27 |
| **Shot Noise** | ID accuracy | -0.73 | 0.2 | 0.26 | 0.18 | 0.9 | **0.96** |
| | Jacobian | -0.02 | 0.3 | -0.0 | 0.06 | -0.0 | 0.07 |
| | Pixel - within | 3.92 | 0.09 | -2.18 | 0.22 | -7.23 | **0.84** |
| | Pixel - between | -12.26 | 0.2 | 1.73 | 0.09 | 7.45 | **0.57** |
| | Amp - HFF | 3.66 | 0.25 | -2.3 | 0.21 | -5.87 | **0.79** |
| | Amp - CD | -0.03 | 0.23 | 0.03 | 0.3 | 0.04 | **0.71** |
| | Phase - HFF | 7.47 | 0.25 | 1.22 | 0.05 | 6.31 | 0.16 |
| | Phase - CD | -0.02 | 0.26 | 0.02 | 0.15 | 0.03 | 0.25 |

## A.9 CLIP results on additional corruptions from ImageNet-C

Figure 12 and Table 9 extend Figure 4 and Table 3 to the remaining corruptions from ImageNet-C.

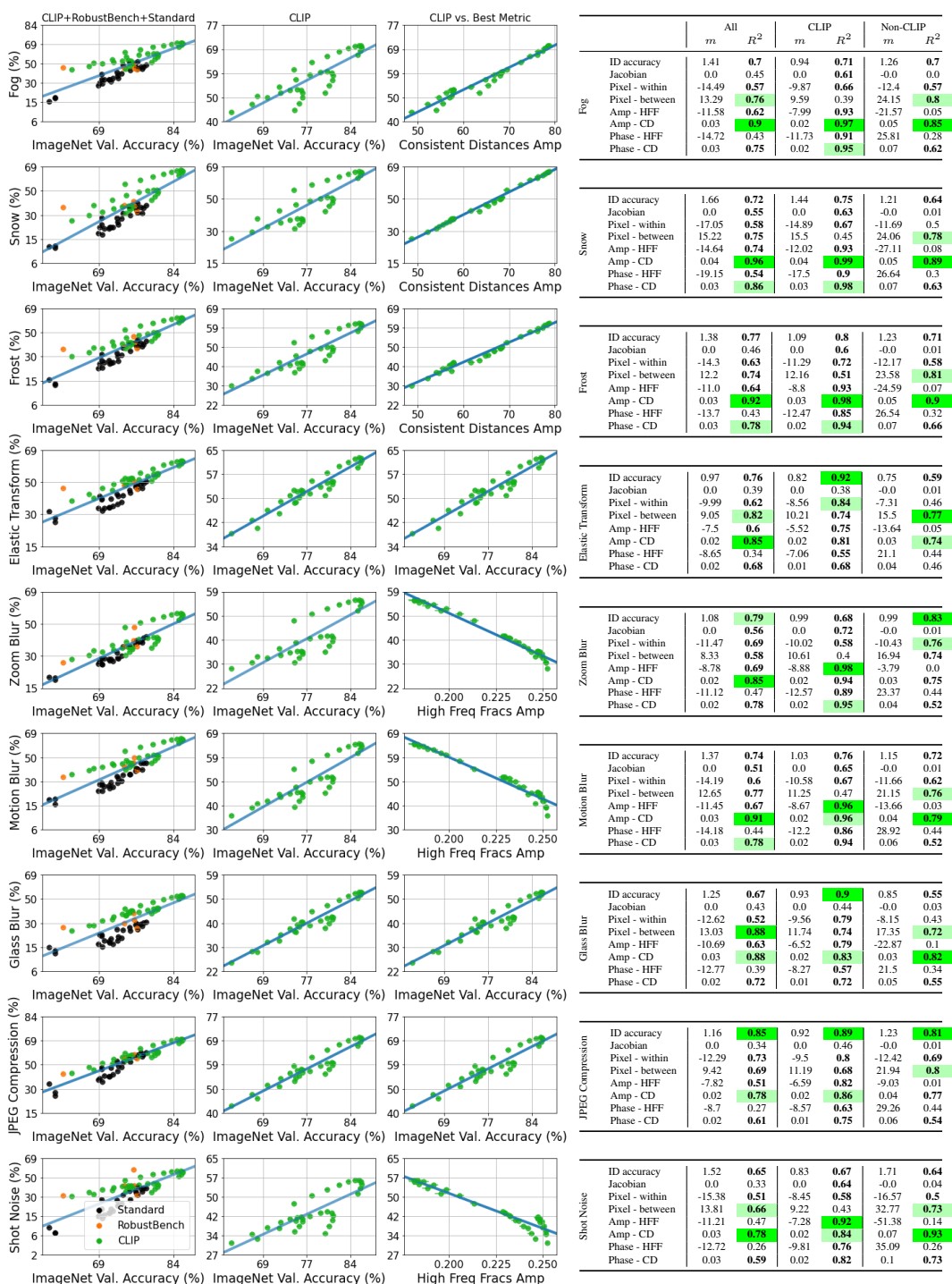

**Fog**

| | All m | All $R^2$ | CLIP m | CLIP $R^2$ | Non-CLIP m | Non-CLIP $R^2$ |
|---|---|---|---|---|---|---|
| ID accuracy | 1.41 | 0.7 | 0.94 | 0.71 | 1.26 | 0.7 |
| Jacobian | 0.0 | 0.45 | 0.0 | 0.61 | -0.0 | 0.0 |
| Pixel - within | -14.49 | 0.57 | -9.87 | 0.66 | -12.4 | 0.57 |
| Pixel - between | 13.29 | 0.76 | 9.59 | 0.39 | 24.15 | 0.8 |
| Amp - HFF | -11.58 | 0.62 | -7.99 | 0.93 | -21.57 | 0.05 |
| Amp - CD | 0.03 | 0.9 | 0.02 | 0.97 | 0.05 | 0.85 |
| Phase - HFF | -14.72 | 0.43 | -11.73 | 0.91 | 25.81 | 0.28 |
| Phase - CD | 0.03 | 0.75 | 0.02 | 0.95 | 0.07 | 0.62 |

**Snow**

| | All m | All $R^2$ | CLIP m | CLIP $R^2$ | Non-CLIP m | Non-CLIP $R^2$ |
|---|---|---|---|---|---|---|
| ID accuracy | 1.66 | 0.72 | 1.44 | 0.75 | 1.21 | 0.64 |
| Jacobian | 0.0 | 0.55 | 0.0 | 0.63 | -0.0 | 0.01 |
| Pixel - within | -17.05 | 0.58 | -14.89 | 0.67 | -11.69 | 0.5 |
| Pixel - between | 15.22 | 0.75 | 15.5 | 0.45 | 24.06 | 0.78 |
| Amp - HFF | -14.64 | 0.74 | -12.02 | 0.93 | -27.11 | 0.08 |
| Amp - CD | 0.04 | 0.96 | 0.04 | 0.99 | 0.05 | 0.89 |
| Phase - HFF | -19.15 | 0.54 | -17.5 | 0.9 | 26.64 | 0.3 |
| Phase - CD | 0.03 | 0.86 | 0.03 | 0.98 | 0.07 | 0.63 |

**Frost**

| | All m | All $R^2$ | CLIP m | CLIP $R^2$ | Non-CLIP m | Non-CLIP $R^2$ |
|---|---|---|---|---|---|---|
| ID accuracy | 1.38 | 0.77 | 1.09 | 0.8 | 1.23 | 0.71 |
| Jacobian | 0.0 | 0.46 | 0.0 | 0.6 | -0.0 | 0.01 |
| Pixel - within | -14.3 | 0.63 | -11.29 | 0.72 | -12.17 | 0.58 |
| Pixel - between | 12.2 | 0.74 | 12.16 | 0.51 | 23.58 | 0.81 |
| Amp - HFF | -11.0 | 0.64 | -8.8 | 0.93 | -24.59 | 0.07 |
| Amp - CD | 0.03 | 0.92 | 0.03 | 0.98 | 0.05 | 0.9 |
| Phase - HFF | -13.7 | 0.43 | -12.47 | 0.85 | 26.54 | 0.32 |
| Phase - CD | 0.03 | 0.78 | 0.02 | 0.94 | 0.07 | 0.66 |

**Elastic Transform**

| | All m | All $R^2$ | CLIP m | CLIP $R^2$ | Non-CLIP m | Non-CLIP $R^2$ |
|---|---|---|---|---|---|---|
| ID accuracy | 0.97 | 0.76 | 0.82 | 0.92 | 0.75 | 0.59 |
| Jacobian | 0.0 | 0.39 | 0.0 | 0.38 | -0.0 | 0.01 |
| Pixel - within | -9.99 | 0.62 | -8.56 | 0.84 | -7.31 | 0.46 |
| Pixel - between | 9.05 | 0.82 | 10.21 | 0.74 | 15.5 | 0.77 |
| Amp - HFF | -7.5 | 0.6 | -5.52 | 0.75 | -13.64 | 0.05 |
| Amp - CD | 0.02 | 0.85 | 0.02 | 0.81 | 0.03 | 0.74 |
| Phase - HFF | -8.65 | 0.34 | -7.06 | 0.55 | 21.1 | 0.44 |
| Phase - CD | 0.02 | 0.68 | 0.01 | 0.68 | 0.04 | 0.46 |

**Zoom Blur**

| | All m | All $R^2$ | CLIP m | CLIP $R^2$ | Non-CLIP m | Non-CLIP $R^2$ |
|---|---|---|---|---|---|---|
| ID accuracy | 1.08 | 0.79 | 0.99 | 0.68 | 0.99 | 0.83 |
| Jacobian | 0.0 | 0.56 | 0.0 | 0.72 | -0.0 | 0.01 |
| Pixel - within | -11.47 | 0.69 | -10.02 | 0.58 | -10.43 | 0.76 |
| Pixel - between | 8.33 | 0.58 | 10.61 | 0.4 | 16.94 | 0.74 |
| Amp - HFF | -8.78 | 0.69 | -8.88 | 0.98 | -3.79 | 0.0 |
| Amp - CD | 0.02 | 0.85 | 0.02 | 0.94 | 0.03 | 0.75 |
| Phase - HFF | -11.12 | 0.47 | -12.57 | 0.89 | 23.37 | 0.44 |
| Phase - CD | 0.02 | 0.78 | 0.02 | 0.95 | 0.04 | 0.52 |

**Motion Blur**

| | All m | All $R^2$ | CLIP m | CLIP $R^2$ | Non-CLIP m | Non-CLIP $R^2$ |
|---|---|---|---|---|---|---|
| ID accuracy | 1.37 | 0.74 | 1.03 | 0.76 | 1.15 | 0.72 |
| Jacobian | 0.0 | 0.51 | 0.0 | 0.65 | -0.0 | 0.01 |
| Pixel - within | -14.19 | 0.6 | -10.58 | 0.67 | -11.66 | 0.62 |
| Pixel - between | 12.65 | 0.77 | 11.25 | 0.47 | 21.15 | 0.76 |
| Amp - HFF | -11.45 | 0.67 | -8.67 | 0.96 | -13.66 | 0.03 |
| Amp - CD | 0.03 | 0.91 | 0.02 | 0.96 | 0.04 | 0.79 |
| Phase - HFF | -14.18 | 0.44 | -12.2 | 0.86 | 28.92 | 0.44 |
| Phase - CD | 0.03 | 0.78 | 0.02 | 0.94 | 0.06 | 0.52 |

**Glass Blur**

| | All m | All $R^2$ | CLIP m | CLIP $R^2$ | Non-CLIP m | Non-CLIP $R^2$ |
|---|---|---|---|---|---|---|
| ID accuracy | 1.25 | 0.67 | 0.93 | 0.9 | 0.85 | 0.55 |
| Jacobian | 0.0 | 0.43 | 0.0 | 0.44 | -0.0 | 0.03 |
| Pixel - within | -12.62 | 0.52 | -9.56 | 0.79 | -8.15 | 0.43 |
| Pixel - between | 13.03 | 0.88 | 11.74 | 0.74 | 17.35 | 0.72 |
| Amp - HFF | -10.69 | 0.63 | -6.52 | 0.79 | -22.87 | 0.1 |
| Amp - CD | 0.03 | 0.88 | 0.02 | 0.83 | 0.03 | 0.82 |
| Phase - HFF | -12.77 | 0.39 | -8.27 | 0.57 | 21.5 | 0.34 |
| Phase - CD | 0.02 | 0.72 | 0.01 | 0.72 | 0.05 | 0.55 |

**JPEG Compression**

| | All m | All $R^2$ | CLIP m | CLIP $R^2$ | Non-CLIP m | Non-CLIP $R^2$ |
|---|---|---|---|---|---|---|
| ID accuracy | 1.16 | 0.85 | 0.92 | 0.89 | 1.23 | 0.81 |
| Jacobian | 0.0 | 0.34 | 0.0 | 0.46 | -0.0 | 0.01 |
| Pixel - within | -12.29 | 0.73 | -9.5 | 0.8 | -12.42 | 0.69 |
| Pixel - between | 9.42 | 0.69 | 11.19 | 0.68 | 21.94 | 0.8 |
| Amp - HFF | -7.82 | 0.51 | -6.59 | 0.82 | -9.03 | 0.01 |
| Amp - CD | 0.02 | 0.78 | 0.02 | 0.86 | 0.04 | 0.77 |
| Phase - HFF | -8.7 | 0.27 | -8.57 | 0.63 | 29.26 | 0.44 |
| Phase - CD | 0.02 | 0.61 | 0.01 | 0.75 | 0.06 | 0.54 |

**Shot Noise**

| | All m | All $R^2$ | CLIP m | CLIP $R^2$ | Non-CLIP m | Non-CLIP $R^2$ |
|---|---|---|---|---|---|---|
| ID accuracy | 1.52 | 0.65 | 0.83 | 0.67 | 1.71 | 0.64 |
| Jacobian | 0.0 | 0.33 | 0.0 | 0.64 | -0.0 | 0.04 |
| Pixel - within | -15.38 | 0.51 | -8.45 | 0.58 | -16.57 | 0.5 |
| Pixel - between | 13.81 | 0.66 | 9.22 | 0.43 | 32.77 | 0.73 |
| Amp - HFF | -11.21 | 0.47 | -7.28 | 0.92 | -51.38 | 0.14 |
| Amp - CD | 0.03 | 0.78 | 0.02 | 0.84 | 0.07 | 0.93 |
| Phase - HFF | -12.72 | 0.26 | -9.81 | 0.76 | 35.09 | 0.26 |
| Phase - CD | 0.03 | 0.59 | 0.02 | 0.82 | 0.1 | 0.73 |

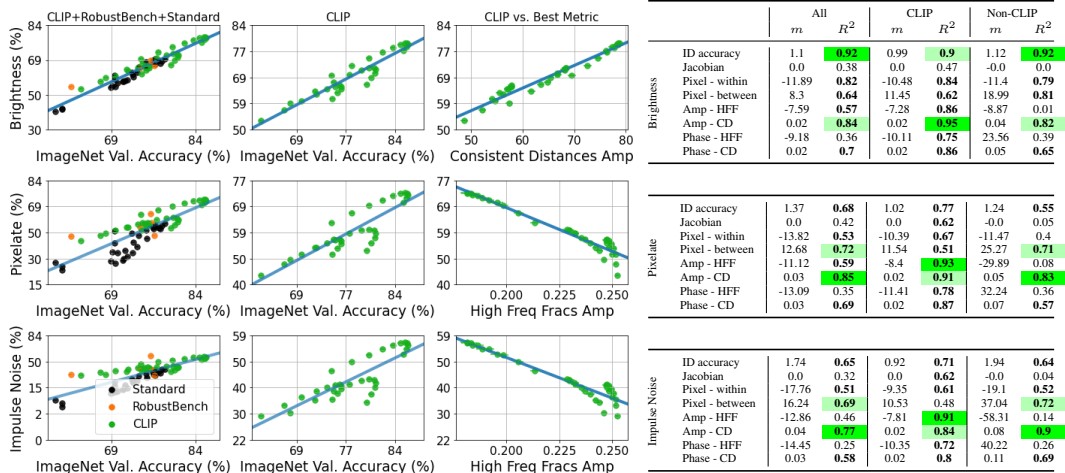

Figure 12: When does CLIP ensembling affect OOD robustness?

Table 9: How well do metrics for CLIP models correlate with OOD robustness?

| | | All | | CLIP | | Non-CLIP | |
|---|---|---|---|---|---|---|---|
| | | $m$ | $R^2$ | $m$ | $R^2$ | $m$ | $R^2$ |
| Brightness | ID accuracy | 1.1 | **0.92** | 0.99 | **0.9** | 1.12 | **0.92** |
| | Jacobian | 0.0 | 0.38 | 0.0 | 0.47 | -0.0 | 0.0 |
| | Pixel - within | -11.89 | **0.82** | -10.48 | **0.84** | -11.4 | **0.79** |
| | Pixel - between | 8.3 | **0.64** | 11.45 | **0.62** | 18.99 | **0.81** |
| | Amp - HFF | -7.59 | **0.57** | -7.28 | **0.86** | -8.87 | 0.01 |
| | Amp - CD | 0.02 | **0.84** | 0.02 | **0.95** | 0.04 | **0.82** |
| | Phase - HFF | -9.18 | 0.36 | -10.11 | **0.75** | 23.56 | 0.39 |
| | Phase - CD | 0.02 | **0.7** | 0.02 | **0.86** | 0.05 | **0.65** |
| Pixelate | ID accuracy | 1.37 | **0.68** | 1.02 | **0.77** | 1.24 | **0.55** |
| | Jacobian | 0.0 | 0.42 | 0.0 | **0.62** | -0.0 | 0.4 |
| | Pixel - within | -13.82 | **0.53** | -10.39 | **0.67** | -11.47 | 0.4 |
| | Pixel - between | 12.68 | **0.72** | 11.54 | **0.51** | 25.27 | **0.71** |
| | Amp - HFF | -11.12 | **0.59** | -8.4 | **0.93** | -29.89 | 0.08 |
| | Amp - CD | 0.03 | **0.85** | 0.02 | **0.91** | 0.05 | **0.83** |
| | Phase - HFF | -13.09 | 0.35 | -11.41 | **0.78** | 32.24 | 0.36 |
| | Phase - CD | 0.03 | **0.69** | 0.02 | **0.87** | 0.07 | **0.57** |
| Impulse Noise | ID accuracy | 1.74 | **0.65** | 0.92 | **0.71** | 1.94 | **0.64** |
| | Jacobian | 0.0 | 0.32 | 0.0 | **0.62** | -0.0 | 0.04 |
| | Pixel - within | -17.76 | **0.51** | -9.35 | **0.61** | -19.1 | **0.52** |
| | Pixel - between | 16.24 | **0.69** | 10.53 | 0.48 | 37.04 | **0.72** |
| | Amp - HFF | -12.86 | 0.46 | -7.81 | **0.91** | -58.31 | 0.14 |
| | Amp - CD | 0.04 | **0.77** | 0.02 | **0.84** | 0.08 | **0.9** |
| | Phase - HFF | -14.45 | 0.25 | -10.35 | **0.72** | 40.22 | 0.26 |
| | Phase - CD | 0.03 | **0.58** | 0.02 | **0.8** | 0.11 | **0.69** |

## A.10  ImageNet data augmentation results on all corruptions

Figures 5 and 13 and Tables 4 and 10 visualize the best metric for non-CLIP ImageNet models, including models from RobustBench [5] trained with data augmentation. Specifically, the RobustBench models are ResNet-50 models trained using the following data augmentations: Stylized-ImageNet [14], DeepAugment [22], and AugMix [21]. Importantly, our baseline models include a ResNet-50 trained without special data augmentations, allowing us to see the effect that the aforementioned data augmentations have on the predictability of robustness while holding architecture constant.

Similar to the robustness of CLIP models, the robustness of non-CLIP ImageNet models is often best predicted by our Fourier interpolation metrics (in particular, CD for amplitude interpolation does well in this setting). In-distribution accuracy can also show strong $R^2$, but it apparently has more trouble predicting the robustness of out-of-line models than our metrics. This phenomenon is illustrated on distribution-shifts like Pixelate and Gaussian Noise (see the corresponding rows of Figures 5 and 13), where in-distribution accuracy can predict many of the models' robustnesses but clearly fails on the effectively robust models from RobustBench. Notably, in these cases, our amplitude CD metric can predict these effectively-robust models' performances as well as it can predict the non-effectively-robust models' performances.

Finally, we emphasize that, despite analyzing a variety of architectures, the out-of-line non-CLIP models on most distribution shifts are almost always the RobustBench models. Importantly, this is consistent with the focus on data augmentation for adding robustness [21, 14, 22], but it also suggests that the construction of architectures with effective robustness is a potential direction for future work.

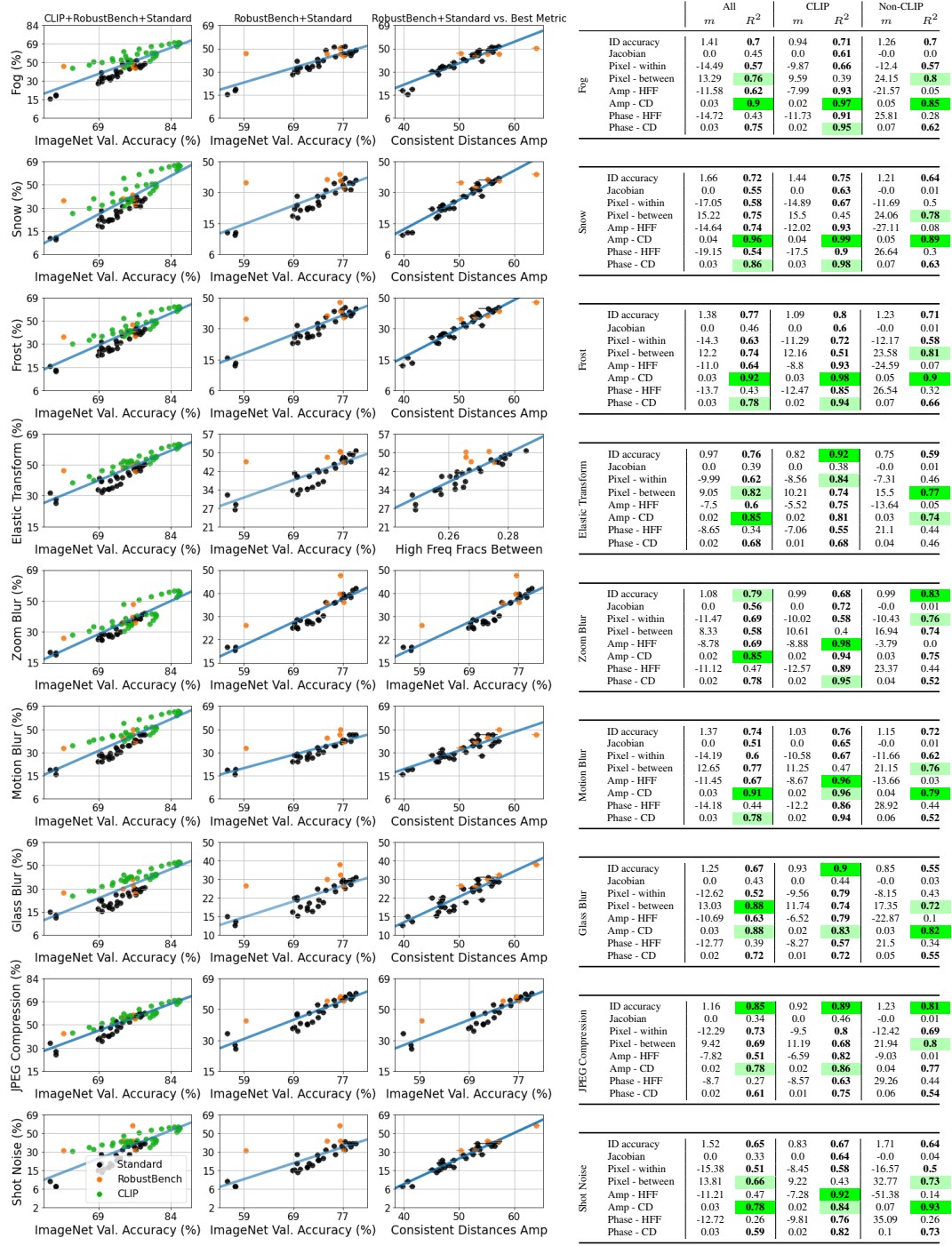

**Fog**

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

Table 10: How well do metrics for ImageNet models correlate with OOD robustness?