# OpenReview forum: "Models Out of Line: A Fourier Lens on Distribution Shift Robustness"
_NeurIPS.cc/2022/Conference — NeurIPS 2022 Accept_

### Official Review · Reviewer_N1xd · 2022-06-26

**Rating:** 6
**Confidence:** 3
**Soundness:** 3 good
**Presentation:** 3 good
**Contribution:** 3 good

**Summary:**

This paper takes metrics which predict OOD robustness (such as ID accuracy, model jacobian norm, etc.) and then looks at the correlation between these metrics and OOD accuracy for methods which can produce robust models. This paper observes that while ID accuracy is a good predictor of robustness for natural distribution shifts (ImageNet -> ImageNetV2), for some synthetic distribution shifts there are better metrics. In general they find "no single metric rules them all". The paper also introduces new metrics which correlate will with the robustness of ensembled CLIP models.

**Questions:**

- Why do the authors think that proposed metrics correlate well for ensembled CLIP models?
- Can any of the proposed metrics be used as objectives to train better models?
- What happens on additional natural distribution shifts such as ImageNetR or ObjectNet or a shift from WILDS?


**Limitations:**

Yes

**Strengths And Weaknesses:**

Strengths:
Overall the results are very interesting and the empirical findings are likely useful to the community. Moreover, the proposed metrics seem that they may be potentially useful in further understanding CLIPs robustness. Finally, robustnets could be a useful resource.

Weaknesses:
My thoughts are that the main weakness of this paper was the lack of distribution shifts, in particular natural distribution shifts which are more likely to occur in the world. I would have appreciated results for additional distribution shifts including ImageNetR or ObjectNet. In addition, it's not clear if there is a takeaway for designing better models. Finally, while a conclusion of this paper is that there is no one metric to rule them all, there is only one column of plots without ID accuracy on the x-axis. Consider showing other metrics on the x-axis in the appendix.

---

> ### Author Response · Authors · 2022-08-02
> **Experiments on additional datasets strengthen results**
>
> Thank you for your positive and constructive feedback. We respond to individual questions/concerns inline below. Please note our summative comments in the joint response.
>
> > My thoughts are that the main weakness of this paper was the lack of distribution shifts, in particular natural distribution shifts which are more likely to occur in the world. I would have appreciated results for additional distribution shifts including ImageNetR or ObjectNet.
>
> In the original manuscript we include results on CIFAR-10.1 and ImageNetV2, both of which are natural distribution shifts. In the revised paper, we have added results on ImageNet-R, ImageNet-A, ImageNet-Sketch, and ObjectNet. We find that our Fourier interpolation metrics continue to produce high correlations (>=0.93 R2) on all of these additional datasets. We thank the reviewer for the suggestion as we believe adding these additional datasets serves to further strengthen and validate our results!
> Please note that we provide more detailed results for these datasets in our response to the related question below. Additionally, we note that not all of these four new datasets contain strictly natural distribution shifts, but they include the datasets requested by the reviewer (i.e., ImageNet-R and ObjectNet) and more.
>
>
> > In addition, it's not clear if there is a takeaway for designing better models.
>
> We have added a discussion to our revised conclusion, recommending that future work consider adapting our Fourier interpolation metrics as regularizers or data augmentation strategies to use during training, as this may improve robustness.
>
> > Finally, while a conclusion of this paper is that there is no one metric to rule them all, there is only one column of plots without ID accuracy on the x-axis. Consider showing other metrics on the x-axis in the appendix.
>
> We agree; thanks for the suggestion! We have added Figures 9 and 10 to the revised appendix to visualize the most predictive metrics on each of the CIFAR-10 distribution shifts, and Figure 12 to visualize the most predictive metrics on each of the ImageNet distribution shifts for non-CLIP models.
>
> > Why do the authors think that proposed metrics correlate well for ensembled CLIP models?
>
> Our experiments show that CLIP models achieve robustness precisely when they exhibit insensitivity to perturbations of Fourier amplitude (and sometimes phase). We designed these Fourier interpolation metrics to align with our human notion of visual semantic meaning, since human viewers are insensitive to perturbations of Fourier amplitude. It is tempting to believe that CLIP models correlate well with these metrics because they share a similar notion of semantic meaning, perhaps due to the way they were trained alongside natural language, but this hypothesis remains to be tested by future research. Our goal in this paper is to show that CLIP robustness does indeed correlate well with our proposed metrics.
>
> > Can any of the proposed metrics be used as objectives to train better models?
>
> Possibly, and we suggest this in our revised conclusions section. The aim of our paper is to explore and compare candidate metrics that might help explain model robustness, a necessary first step that we hope will be followed up by future work that adapts our metrics into regularizers or augmentation strategies for training more robust models.
>
> > What happens on additional natural distribution shifts such as ImageNetR or ObjectNet or a shift from WILDS?
>
> We thank the reviewer for the question! Our revised manuscript shows that our metrics are particularly illuminating with respect to CLIP performance on these datasets: On ImageNet-R, one of our newly introduced metrics attains 0.93 R2; in-distribution accuracy attains 0.31 R2. On ObjectNet, two of our newly introduced metrics attain 0.96 R2; in-distribution accuracy attains 0.48 R2. On ImageNet-Sketch, two of our proposed metrics attain 0.95 R2; in-distribution accuracy attains 0.52 R2.  On ImageNet-A, one of our proposed metrics attains 0.97 R2; in-distribution accuracy attains 0.42 R2. These results along with R2s for non-CLIP models are visible in Figure 4 and Table 3 of the revised manuscript (also, the most predictive metrics for non-CLIP models are visualized in Figure 12). Again, we note that not all of these datasets contain strictly natural distribution shifts, but they include the datasets requested by the reviewer (i.e., ImageNet-R and ObjectNet) and more.
>
> We did not evaluate on the WILDS distribution shifts because, to the best of our knowledge, the literature has not yet shown models with effective robustness on most of these more challenging shifts (see https://wilds.stanford.edu/leaderboard/ for current results on WILDS).

---

> > ### Comment · Reviewer_N1xd · 2022-08-04
> > **Thanks for the response.**
> >
> > Thank you to the authors for their response. My concerns are addressed and I continue to recommend acceptance.

---

### Official Review · Reviewer_YyCi · 2022-07-10

**Rating:** 6
**Confidence:** 4
**Soundness:** 2 fair
**Presentation:** 3 good
**Contribution:** 2 fair

**Summary:**

This paper carries out a comprehensive study of the Distribution Shift Robustness through a Fourier lens and designs new 'Fourier sensitivity metrics' that capture Fourier sensitivity of models as test data moves farther away from the training data manifold. It provides the theoretical analysis of the OOD robustness on pruning, data augmentation, and weight ensembling.

**Questions:**

The questions are listed in the pros and cons.

**Limitations:**

N.A.

**Strengths And Weaknesses:**

Strength:
(1) This paper considers the OOD robustness on pruning, data augmentation, and weight ensembling.
(2) This paper designs new metrics that capture the Fourier sensitivity of models.

Weaknesses:
(1) To the best of the reviewer's knowledge, this is one existing work [1*] considering both the amplitude and phase on the robustness in the frequency, while [1*] is not in this paper's reference list. The design difference is needed, at least.

(2) The Power spectral densities in Figure 2 are nearly the same as those in [36]. A citation of [36] is needed in Sec. 4.1.

(3)  The experiment setting is a little strange. The experiments in Secs. 4.2 and 4.3 are conducted on CIFAR-10, while ImageNet is used in Sec. 4.4. From the reviewer's view, it is better to conduct the experiments on one consistent dataset or on both datasets.

(4) Since all the experiments are conducted on corruption benchmarks (CIFAR10-C and ImageNet-C), the separated experimental results on each type of corruption are suggested to add.

[1*] Amplitude-Phase Recombination: Rethinking Robustness of Convolutional Neural Networks in Frequency Domain. ICCV 2021


-------------------------------------------------------
Post-rebuttal reviews:
My main concerns are on the experimental settings, which have been well-addressed in the rebuttal. I will raise the recommendation to 'weak accept'.

---

> ### Author Response · Authors · 2022-08-02
> **Clarification of our contributions and our experimental settings 2/2**
>
> > The experiment setting is a little strange. The experiments in Secs. 4.2 and 4.3 are conducted on CIFAR-10, while ImageNet is used in Sec. 4.4. From the reviewer's view, it is better to conduct the experiments on one consistent dataset or on both datasets.
>
> We respond in the joint response, reproduced here for convenience:
>
> Thank you for raising this point! Our revision clarifies that all of our experiments are driven by the need to study effectively robust or “out of line” models, which are quite rare (see [1]) but can be found among CIFAR-10 and ImageNet models. For CIFAR-10, these out-of-line models have been produced via model pruning [2] and data augmentation [3], so these are the settings we consider on that dataset in Figures 2, 3, 7–10 and Tables 1, 2, 4, 5. For ImageNet, out-of-line models have been produced via CLIP fine-tuning [4] and data augmentation [3], so we consider these settings in Figures 4, 11, 12 and Tables 3, 8, 9. To enhance our presentation’s consistency across these settings, our revision now includes visualizations of the most predictive metrics for each setting, whereas before this information was only visible for CLIP models—e.g., Figure 12 plots the metric most predictive of robustness for models trained on ImageNet with and without data augmentation and Figure 9 plots the metric most predictive of pruned CIFAR-10 model robustness. We emphasize that out-of-line models are difficult to find, and that our experiments are carefully selected to increase our understanding of robustness in these rare settings where it has been shown to occur.
>
> [1] Andreassen, A., Bahri, Y., Neyshabur, B., & Roelofs, R. (2021). The evolution of out-of-distribution robustness throughout fine-tuning. arXiv preprint arXiv:2106.15831.
> [2] Diffenderfer, J., Bartoldson, B., Chaganti, S., Zhang, J., & Kailkhura, B. (2021). A winning hand: Compressing deep networks can improve out-of-distribution robustness. Advances in Neural Information Processing Systems, 34, 664-676.
> [3] Croce, F., Andriushchenko, M., Sehwag, V., Debenedetti, E., Flammarion, N., Chiang, M., Mittal, P., & Hein, M. (2020). Robustbench: a standardized adversarial robustness benchmark. arXiv preprint arXiv:2010.09670.
> [4] Wortsman, M., Ilharco, G., Kim, J. W., Li, M., Kornblith, S., Roelofs, R., Gontijo-Lopes, R., Hajishirzi, H., Farhadi, A., Namkoong, H., & Schmidt, L. (2022). Robust fine-tuning of zero-shot models. In Proceedings of the IEEE/CVF Conference on Computer Vision and Pattern Recognition (pp. 7959-7971).
>
> > Since all the experiments are conducted on corruption benchmarks (CIFAR10-C and ImageNet-C), the separated experimental results on each type of corruption are suggested to add.
>
> Please note that these results are already in the original submission, with representative corruptions (spread among low, medium, and high frequencies) in the main text figures and the remaining corruptions in the appendix (each visualized individually). As stated in the text, the trends reflected in the main text results hold in the appendix results.
>
> It is also important to note that not “all” of the experiments use “corruption benchmarks”. Indeed, many of the results use natural distribution shifts like CIFAR-10.1 and ImageNet V2 in the original submission, and the new submission adds natural distribution shifts via ImageNet-A and ObjectNet (as well as less natural shifts via ImageNet-R and ImageNet-Sketch). Each of these benchmarks is also visualized individually.

---

> > ### Comment · Reviewer_YyCi · 2022-08-08
> > **Thanks for the response.**
> >
> > Thanks a lot for the authors' responses. After seeing the rebuttal and other reviewers' comments and checking the revised submission, I think my main concerns are well-addressed and I will raise the recommendation to 'weak accept'.

---

> ### Author Response · Authors · 2022-08-02
> **Clarification of our contributions and our experimental settings 1/2**
>
> Thank you for your time and effort spent reviewing our work. We believe that your concerns primarily relate to clarity/presentation, and we have addressed each of your concerns inline below. Notably, we wanted to highlight that our focus is on making progress towards identifying and understanding **effective** robustness of **out-of-line models**, which are exceedingly rare. These models behave very differently compared to standard (or on-the-line) models and are believed to be the key to solving the OOD robustness puzzle  (see [a] for a more detailed discussion). In this context, our main contributions (i.e., establishing new model metrics that demonstrate strong correlation with OOD robustness for CLIP models, illustrating nuance in correlation of model metrics with OOD robustness, and an open source collection of pretrained CIFAR-10 models with effective robustness) remain unaffected by your concerns. Furthermore, we carried out new experiments (on 22 new models and 4 new OOD datasets) that show that our findings are more broadly applicable than the cases we originally considered, in turn making our contribution much stronger than initially perceived (please see our joint response). Lastly, we want to clarify that the focus of this work is very different than the focus of [1*], a point which we elaborate on in our inline responses below.
>
> We hope that you find our response to your concerns satisfactory and that you will update your score and champion this work.
>
> [a] Andreassen, A., Bahri, Y., Neyshabur, B., & Roelofs, R. (2021). The evolution of out-of-distribution robustness throughout fine-tuning. arXiv preprint arXiv:2106.15831.
>
> >To the best of the reviewer's knowledge, this is one existing work [1*] considering both the amplitude and phase on the robustness in the frequency, while [1*] is not in this paper's reference list. The design difference is needed, at least.
>
> Thank you for making us aware of this paper! Crucially, our focus is very different compared to [1] in that our goal is predicting the OOD robustness of preexisting **out-of-line models**. In our revision, we cite [1] to highlight that prior work has found that exchanging amplitude information between images can improve robustness. Please note that the design difference is as follows: our work uses a post-training analysis to gradually interpolate between two images’ amplitude (or phase) information to construct probing-images that quantify a trained model’s robustness; separately, [1] boosts robustness via a training-time data augmentation that completely exchanges images’ amplitude information rather than interpolating.
>
> [1] Chen, G., Peng, P., Ma, L., Li, J., Du, L., & Tian, Y. (2021). Amplitude-phase recombination: Rethinking robustness of convolutional neural networks in frequency domain. In Proceedings of the IEEE/CVF International Conference on Computer Vision (pp. 458-467).
>
> > The Power spectral densities in Figure 2 are nearly the same as those in [36]. A citation of [36] is needed in Sec. 4.1.
>
> We have added a citation to [36] in Section 4.1, which has been merged with appendix Section A.5 in the revision (due to space constraints). Please note that these PSDs themselves are not claimed to be a contribution of our paper; however, the PSD analysis of CIFAR-10.1 is novel (never before visualized) and required the approach described in Section A.5. More importantly, these PSDs are connected to our claim that no one metric rules them all. Indeed, Section A.5 clarifies that the reason we include these PSDs is to visualize the spectral properties of the distribution shifts we study on CIFAR-10, which can foster intuition for why “each distribution shift is unique” and “has its own problem structure” (RxwV).

---

### Official Review · Reviewer_RxwV · 2022-07-11

**Rating:** 7
**Confidence:** 4
**Soundness:** 3 good
**Presentation:** 3 good
**Contribution:** 3 good

**Summary:**

**High level goal:** The authors are interested in an empirical study of models that display *effective robustness* (those that are "above the line" that's traced by in and out-of-distribution accuracies of models that don't display effective robustness) in the image domain.

**Gap in literature that the paper contributes in addressing:** The conditions (on OOD data, model and training techniques) under which effective generalization is achieved is not well understood.

**Main contributions:**
  * Using previously proposed metrics that are known to correlate with OOD generalization (including some new metrics based on the Fourier spectra of the images and models), the authors study how different types of pruning, data augmentation and ensembling affect effective generalization.
  * The authors release the RobustNets suite of models that display effective robustness to facilitate further research in this space.
  * The Fourier-based metrics proposed by the authors correlate well with effective generalization of models form the CLIP family. It's also interesting that these metrics don't correlate particularly strongly when other (mostly convnet based) models are used, which is useful in gaining indirect knowledge about the types of functions represented by these networks.







**Questions:**

* Could you explain which models you evaluated on ImageNet? The description (between lines 109 and 112) is a bit vague. Which architectures are included, for example?
* Line 153: Could you clarify what you mean by "we interpolate the lowest 40% of image frequencies"? Are these the frequencies that occupy 40% of the energy? Or perhaps something else? Also, do you leave the rest of the frequencies untouched?
* Line 280: Are the 11 CLIP models obtained by interpolating the same pretrained and finetuned versions of CLIP?


Nitpick: It perhaps could be interesting to add a plot in the Appendix that supplements Figure 1 in that instead of tracing amplitude or phase interpolations along a single axis, a grid of images can be provided where on the left top and right bottom, we have the unperturbed images, and in the middle we have various (mixed) interpolations.

**Limitations:**

**Only image-based distribution shifts:** The proposed metrics (and the analysis) is only concerned with image data.
**Experiments have no causal power:** Despite authors' claims, the experiments are fully correlational, and this should be acknowledged so. (see weaknesses for above)


**Strengths And Weaknesses:**

STRENGTHS:
* **Relevance:** The focus of the paper is well within the scope of NeurIPS, especially given the recent interest in gaining a deeper understanding of OOD generalization.
* **Comprehensive evaluation:** The experiments are quite extensive, providing a plethora of useful data for understanding the conditions under which effective generalization occurs.
* **Usefulness of the spectral characteristics of in and OOD data:** It turns out that the spectral characteristics of in and OOD data gives useful information regarding on which distributions shifts are more likely to induce "accuracy-on-the-line' style phenomenon.
* **Evidence that each distribution shift is "unique"** The authors make a compelling case that each distribution shift has it's own problem structure, and there's not single metric that correlates well with generalization across all shifts (at least among the ones explored in the paper)


WEAKNESS:
* **Lack of causal explanations:** Even though the authors use words that imply the results provide a causal understanding of why effective generalization happens, all of the results are correlational. It's misleading to claim that the experiments can answer "why" questions. I believe it's very important that this is fixed before the camera-ready version, should the paper be accepted. (I'm happy to hear arguments against this criticism) In my opinion, this is an important limitation that can be fixed simply by removing any causal claims from the paper - the existing results are still worthwhile.
* **No emphasis on architecture:** It'd be interesting to see a discussion of whether model architecture has any impact on effective generalization. The authors likely have enough empirical data to address this - a discussion of this would have made the paper stronger.
**Lack of mechanistic understanding:** Despite a plethora of empirical data, I don't think I've acquired a deeper (i.e. mechanistic) understanding of why/how effective robustness occurs. I also am not sure if a practitioner also leaves the paper with a solid understanding of which techniques to use to achieve effective robustness, beyond using methods from already existing literature.
  * This is not to say that the presented results are non meaningful - perhaps follow-up works will fill these gaps.

================
**Post rebuttal update**
I thank the authors for their response. Since some of my important concerns are addressed, which I've reflected on my review by increasing the score.

---

> ### Author Response · Authors · 2022-08-02
> **We have improved clarity based on your comments 2/2**
>
> > Could you explain which models you evaluated on ImageNet? The description (between lines 109 and 112) is a bit vague. Which architectures are included, for example?
>
> Thank you for raising this point! We have added the full list of models to the revised paper (in Appendix A.1).
> - We use the following models from Torchvision (details at https://pytorch.org/vision/0.8/models.html): alexnet, vgg11, vgg11_bn, vgg13, vgg13_bn, vgg16, vgg16_bn, vgg19, vgg19_bn, resnet18, resnet34, resnet50, resnet101, resnet152, squeezenet1_0, squeezenet1_1, densenet121, densenet169 ,densenet161, densenet201, googlenet, shufflenet_v2_x1_0, mobilenet_v2, resnext50_32x4d, resnext101_32x8d, wide_resnet50_2, wide_resnet101_2, and mnasnet1_0.
> - We use the following models from the RobustBench model zoo on ImageNet corruptions (details at https://github.com/RobustBench/robustbench#model-zoo):  Geirhos2018_SIN, Geirhos2018_SIN_IN, Geirhos2018_SIN_IN_IN, Hendrycks2020Many, and Hendrycks2020AugMix.
> - From the CLIP family, we use 33 models formed by weight-space interpolation between zero-shot and fine-tuned ViT-B/16, ViT-B/32, and ViT-L/14 (11 models from each architecture).
>
> > Line 153: Could you clarify what you mean by "we interpolate the lowest 40% of image frequencies"? Are these the frequencies that occupy 40% of the energy? Or perhaps something else? Also, do you leave the rest of the frequencies untouched?
>
> We clarify in our revision's Appendix A.3 that the 40% is with respect to the maximum spatial frequency of the image (a function of the pixel resolution), not based on the actual power spectral density of the image. The interpolation is done using a square mask on the DFT of the image, which interpolates the lower image frequencies but leaves the higher image frequencies unperturbed. We chose these frequency cutoffs based on visualizing the resulting amplitude and phase interpolating paths and visually verifying that the phase paths destroy semantic content (including frequencies that are too high in the phase interpolation can introduce semantically meaningful content from the other/non-original image) and the amplitude paths preserve semantic content (including frequencies that are too high in the amplitude interpolation can produce images that are a bit too corrupted for a human to confidently classify).
>
> > Line 280: Are the 11 CLIP models obtained by interpolating the same pretrained and finetuned versions of CLIP?
>
> Yes, all 11 models are CLIP ViT-B/16, interpolating between zero-shot and fine-tuned in increments of 10%. However, we have updated our paper to include additional results (using the same weight interpolation procedure) on two more CLIP architectures, ViT-B/32 and ViT-L/14, bringing the total number of CLIP models we evaluate to 33. Thank you for the question—we believe our results are strengthened by the addition of these models.
>
> > Nitpick: It perhaps could be interesting to add a plot in the Appendix that supplements Figure 1 in that instead of tracing amplitude or phase interpolations along a single axis, a grid of images can be provided where on the left top and right bottom, we have the unperturbed images, and in the middle we have various (mixed) interpolations.
>
> Thank you for the suggestion! Although this might be a visually interesting figure, the various mixed interpolations you suggest are not utilized by our methodology and thus we are concerned they may create confusion. We would be happy to add this figure, however, if you could please provide a context or motivation for its inclusion.
>
> > Limitations: Only image-based distribution shifts: The proposed metrics (and the analysis) is only concerned with image data. Experiments have no causal power: Despite authors' claims, the experiments are fully correlational, and this should be acknowledged so. (see weaknesses for above)
>
> Our revision’s discussion clarifies that our focus on image data is a limitation of both our work and the broader OOD robustness literature. In particular, we focus on images because there are very few non-image domains in which effective robustness has been identified in the literature. However, prior work has shown NLP embeddings have intuitive spectral properties [1], suggesting that future work may find that our Fourier approach to creating semantics-preserving corruptions can be modified to provide insight when using other data modalities.
>
> We address the “why”/causal limitation above, and in the revised paper.
>
> [1] Tamkin, A., Jurafsky, D., & Goodman, N. (2020). Language through a prism: A spectral approach for multiscale language representations. Advances in Neural Information Processing Systems, 33, 5492-5504.

---

> ### Author Response · Authors · 2022-08-02
> **We have improved clarity based on your comments 1/2**
>
> Thank you for your positive and constructive feedback. We respond to individual questions/concerns inline below. Please note our summative comments in the joint response.
>
> > Lack of causal explanations: Even though the authors use words that imply the results provide a causal understanding of why effective generalization happens, all of the results are correlational. It's misleading to claim that the experiments can answer "why" questions. I believe it's very important that this is fixed before the camera-ready version, should the paper be accepted. (I'm happy to hear arguments against this criticism) In my opinion, this is an important limitation that can be fixed simply by removing any causal claims from the paper - the existing results are still worthwhile.
>
> We agree! Please note that we have revised our discussion (particularly in the “Results” section) to reflect that our experiments provide correlational evidence for and against various hypotheses regarding why effective robustness might emerge. For example, the first paragraph of Section 4 now mentions that establishing causality requires rigorous analysis (as exemplified by the movement from empirical findings to causality found in generalization analyses going from Jiang et al. [18] to Dziugaite and Drouin et al., NeurIPS 2020 https://arxiv.org/pdf/2010.11924.pdf). Importantly, this discussion makes clear that our contribution of providing models and candidate metrics may inform this future work on causality in the OOD/ER space.
>
> > No emphasis on architecture: It'd be interesting to see a discussion of whether model architecture has any impact on effective generalization. The authors likely have enough empirical data to address this - a discussion of this would have made the paper stronger.
>
> Thank you for the suggestion! Architecture-specific trendlines are shown in the CIFAR-10 figures, and we have added a related discussion in the text (please see the revision’s Section 4.1). In most cases, the different architectures exhibit similar trends, but sometimes they differ or are simply offset from each other: e.g., Conv8 is often more “out-of-line” than the other architectures. On ImageNet, we include a range of model architectures in the standard models and find that they are mostly non-robust, and it is the training data that influences out-of-line robustness more than it is the architecture (e.g., please see the discussion of ResNet-50s trained with and without data augmentation in our revision’s Appendix A10). Importantly, we mention this as a worthwhile direction for further, future work, which may become especially relevant as more robust architectures are introduced.
>
> > Lack of mechanistic understanding: Despite a plethora of empirical data, I don't think I've acquired a deeper (i.e. mechanistic) understanding of why/how effective robustness occurs. I also am not sure if a practitioner also leaves the paper with a solid understanding of which techniques to use to achieve effective robustness, beyond using methods from already existing literature… This is not to say that the presented results are non meaningful - perhaps follow-up works will fill these gaps.
>
> Thank you for this comment! In our revised paper, we suggest that future researchers might use our metrics, results, and released models to develop a causal understanding of ER as well as practical methodologies to induce it, for example by applying our Fourier interpolation metrics as regularizers during training. Regarding a mechanistic understanding, effective robustness has only been recently identified in the literature, and there has been slow progress in terms of explaining why ER occurs and designing procedures to induce ER. However, our results and Figure 1 suggest that alignment with human perceptions of semantic meaning in images may be important to effective robustness. In particular, we show that models that are less sensitive to Fourier amplitude interpolation (and sometimes also Fourier phase interpolation) also tend to be more robust across a wide variety of natural and synthetic distribution shifts.

---

### Author Response · Authors · 2022-08-02
**Joint Response 2/2**

- Finally, as noted by Reviewer RxwV, our experiments are correlational rather than causal. Our revision clarifies that our goal is to compare candidate robustness metrics and provide experimental evidence as to which are most promising for future work (e.g. to determine causality and design training improvements). While we fully agree that our present results are correlational only, we do believe that our aforementioned strong results on new datasets suggested by Reviewer N1xd (i.e., ImageNet-A, ImageNet-R, ImageNet-Sketch, and ObjectNet) are a good indicator that our newly-introduced Fourier interpolation metrics merit further research, and that our contribution of providing models (RobustNets) and an “extensive/comprehensive empirical study” (RxwV, YyCi) is critical to these next steps.

We respond to each reviewer’s specific comments and questions individually below. Through these answers and clarifications, we believe we address the concerns of each of the reviewers, highlight the significance of our findings, and convey the crucial role RobustNets and our Fourier interpolation metrics can play for future research addressing the OOD robustness puzzle. We are hopeful that reviewers will consider our answers, increase their ratings, and recommend acceptance. Please feel free to follow up with any additional questions.

Thank you,

Paper3764 Authors

[1] Andreassen, A., Bahri, Y., Neyshabur, B., & Roelofs, R. (2021). The evolution of out-of-distribution robustness throughout fine-tuning. arXiv preprint arXiv:2106.15831.

[2] Diffenderfer, J., Bartoldson, B., Chaganti, S., Zhang, J., & Kailkhura, B. (2021). A winning hand: Compressing deep networks can improve out-of-distribution robustness. Advances in Neural Information Processing Systems, 34, 664-676.

[3] Croce, F., Andriushchenko, M., Sehwag, V., Debenedetti, E., Flammarion, N., Chiang, M., Mittal, P., & Hein, M. (2020). Robustbench: a standardized adversarial robustness benchmark. arXiv preprint arXiv:2010.09670.

[4] Wortsman, M., Ilharco, G., Kim, J. W., Li, M., Kornblith, S., Roelofs, R., Gontijo-Lopes, R., Hajishirzi, H., Farhadi, A., Namkoong, H., & Schmidt, L. (2022). Robust fine-tuning of zero-shot models. In Proceedings of the IEEE/CVF Conference on Computer Vision and Pattern Recognition (pp. 7959-7971).

---

### Author Response · Authors · 2022-08-02
**Joint Response 1/2**

We deeply appreciate the reviewers’ time and thoughtful feedback on our work. We are happy to hear that reviewers found our study of the effective robustness puzzle to be “extensive” (RxwV), “comprehensive” (YyCi), “very interesting” (N1xd), and “likely useful” (N1xd, RxwV). Importantly, reviewers recognized the “compelling case” (RxwV) for our proposed metrics that “capture the Fourier sensitivity” (YyCi) of models and “may be potentially useful in further understanding CLIP’s robustness” (N1xd).

Relatedly, questions raised by reviewers led to our findings being more broadly applicable than the cases we originally considered. In the revised paper, we analyze 33 CLIP models (11 for each of ViT-B/16, ViT-B/32, and ViT-L/14) rather than the 11 (ViT-B/16) in the original submission. We show that our CLIP findings extend to 4 new out-of-distribution datasets for ImageNet (based on N1xd’s suggestion). Specifically, using our proposed Fourier interpolation metrics we are able to predict the robustness of 33 CLIP models on **ImageNet-A**, **ImageNet-Sketch**, **ImageNet-R**, and **ObjectNet** with **0.97**, **0.95**, **0.93**, and **0.96** R2, respectively.

Without our metrics and relying on in-distribution validation accuracy as a predictor, the R2s on these datasets drop to 0.42, 0.52, 0.31, and 0.48, respectively. This gap in correlation quality underscores the difficulty of predicting OOD robustness and the value added by our new metrics. Critically, these results are in agreement with the results we presented in our original submission, which showed our metrics “correlate well with effective [robustness] of models from the CLIP family” (RxwV); without these metrics, practitioners are left without reliable means to predict the robustness of “out of line” models like CLIP.

Notably, reviewers also highlighted the potential usefulness (N1xd) of RobustNets, the set of out-of-line CIFAR-10 models we release to "facilitate further research" (RxwV) into effective robustness. Our study finds these models may be particularly useful in the setting of pruned CIFAR-10 models, which can exhibit effective robustness but remain poorly understood.

However, reviewers appropriately noted areas where our communication of results could be improved, and our revised manuscript carefully addresses these areas:

- Noting that our “experimental setting is a little strange”, Reviewer YyCi suggested that we use “one consistent dataset or both datasets” for each experiment. Our revision clarifies that all of our experiments are driven by the need to study effectively robust or “out of line” models, which are quite rare (see [1]) but can be found among CIFAR-10 and ImageNet models. For CIFAR-10, these out-of-line models have been produced via model pruning [2] and data augmentation [3], so these are the settings we consider on that dataset in Figures 2, 3, 7–10 and Tables 1, 2, 4, 5. For ImageNet, out-of-line models have been produced via CLIP fine-tuning [4] and data augmentation [3], so we consider these settings in Figures 4, 11, 12 and Tables 3, 8, 9. To enhance our presentation’s consistency across these settings, our revision now includes visualizations of the most predictive metrics for each setting, whereas before this information was only visible for CLIP models—e.g., Figure 12 plots the metric most predictive of robustness for models trained on ImageNet with and without data augmentation and Figure 9 plots the metric most predictive of pruned CIFAR-10 model robustness. We emphasize that out-of-line models are difficult to find, and that our experiments are carefully selected to increase our understanding of robustness in these rare settings where it has been shown to occur.

- Relatedly, reviewer N1xd correctly points out that “there is only one column of plots without ID accuracy on the x-axis.” In line with their suggestion, the revised paper shows “other metrics on the x-axis in the appendix” for the CIFAR-10 models and non-CLIP ImageNet models (again, previously we only showed the most predictive metric for CLIP ImageNet models). We agree with the reviewer that these new figures (9, 10, and 12 in our revision’s appendix) help illustrate our findings that there is “no one metric to rule them all”; notably, these new figures also help illustrate that our proposed Fourier interpolation metrics often (particularly on ImageNet) correlate significantly better with robustness than previously-proposed metrics.

---

### Author Response · Authors · 2022-08-08
**Discussion period closing**

Thank you all for your time and thoughtful reviews! As the discussion period comes to a close, we ask that reviewers consider the final, following points of discussion.

**Reviewer RxwV**
Thank you for your positive assessment! We think the clarifications and additional analyses/discussions you suggested have greatly improved our submission. Also, we are grateful for your raising your score from 6 to 7!

**Reviewer N1xd**
Thank you for your positive assessment and manuscript-improving ideas! We hope that we have adequately addressed your main concern about "lack of distribution shifts, in particular natural distribution shifts" with our inclusion of four additional OOD datasets (ImageNet-R, ImageNet-A, ImageNet-Sketch, and ObjectNet). With the corroborating effect of these new experiments on our original conclusions in mind, could you please consider raising your score or letting us know if there is anything else we can address before the author discussion period closes?

**Reviewer YyCi**
As we have not yet received your feedback on our revised paper or on our responses to your original review, could you please consider confirming that we have addressed your concerns and raising your score accordingly? In addition to the clarifying improvements to our manuscript made based on your review, we wish to highlight our revised submission’s inclusion of results for four new OOD datasets for ImageNet (ImageNet-R, ImageNet-A, ImageNet-Sketch, and ObjectNet). We believe these clarifications and corroborating results have strengthened our paper. Additionally, we would be happy to address any new questions you might have in the remaining time. Thank you again for your time and consideration.

Finally, we wish to clarify that, upon acceptance, we would use the extra page available to highlight the new visualizations and analyses suggested by reviewers, which are currently visible in our revised submission’s appendix (e.g., non-CLIP ImageNet results are in Appendix A.10). These additional results further support our take-home message that no one metric rules them all, a surprising conclusion given the previously-demonstrated effectiveness of in-distribution accuracy for predicting robustness in most models [1,2] and the impressive ability of our novel model metrics to predict CLIP and non-CLIP ImageNet model robustness with high accuracy even when in-distribution accuracy fails.

[1] John P Miller, Rohan Taori, Aditi Raghunathan, Shiori Sagawa, Pang Wei Koh, Vaishaal Shankar, Percy Liang, Yair Carmon, and Ludwig Schmidt. Accuracy on the line: on the strong correlation between out-of-distribution and in-distribution generalization. In International Conference on Machine Learning, pages 7721–7735. PMLR, 2021.

[2] Anders Andreassen, Yasaman Bahri, Behnam Neyshabur, and Rebecca Roelofs. The evolution of out-of-distribution robustness throughout fine-tuning, 2021. URL https://arxiv.org/abs/2106.15831.

---

### Meta-Review · Area_Chair_FGcv · 2022-08-23

**Recommendation:** Accept
**Confidence:** Certain

**Metareview:**

All reviewers noted the relevance of the proposed study for the NeurIPS community. They all agreed that the paper is well-motivated, sound, and that the proposed Fourier interpolation is novel. While some reviewers had initial concerns regarding the experimental evaluation, the authors did a great job at improving their experiments and reply to the reviewers's concerns. There, we recommend acceptance.

**Award:**

No

---

### Decision · Program_Chairs · 2022-09-14

Accept